# Symmetries in PAC-Bayesian Learning

**Armin Beck** [1] [2]  **Peter Ochs** [2]

## Abstract

Symmetries are known to improve the empirical performance of machine learning models, yet theoretical guarantees explaining these gains remain limited. Prior work has focused mainly on compact group symmetries and often assumes that the data distribution itself is invariant, an assumption rarely satisfied in real-world applications. In this work, we extend generalization guarantees to the broader setting of non-compact symmetries, such as translations and to non-invariant data distributions. Building on the PAC-Bayes framework, we adapt and tighten existing bounds, demonstrating the approach on McAllester's PAC-Bayes bound while showing that it applies to a wide range of PAC-Bayes bounds. We validate our theory with experiments on several datasets with non-uniform and non-compact transformations, where the derived guarantees not only hold but also improve upon prior results. These findings provide theoretical evidence that, for symmetric data, symmetric models are preferable beyond the narrow setting of compact groups and invariant distributions, opening the way to a more general understanding of symmetries in machine learning.

## 1. Introduction

Many real-world machine learning tasks exhibit inherent symmetric structures, meaning that certain transformations of the input do not change the output. For instance, in image classification, semantic labels remain unchanged under transformations such as rotations, reflections, or translations. Models that explicitly capture such symmetries usually outperform those that do not, as demonstrated empirically in prior work (Cohen & Welling, 2016). Early theoretical frameworks that have analyzed the advantages of incorporating symmetry into models, including (Lyle et al., 2020)

and (Elesedy & Zaidi, 2021), typically relying on two key assumptions: (i) the symmetry group is compact and (ii) the data distributions that are invariant under the group actions.

However, both assumptions are restrictive in many practical scenarios. Real-world data often exhibit more complex or non-compact symmetries. For example, (i) the non-compact group of translations is an important class of symmetries that plays a central role in convolutional architectures of neural networks (Biscione & Bowers, 2021). Furthermore, (ii) real-world data distributions are usually not invariant under relevant transformations, for example, an upside-down filled cup of water is unlikely to occur in natural data.

In this work, we address this gap by studying learning under general non-compact and non-uniform symmetries. Our analysis is based on a PAC-Bayesian framework, which is particularly well suited to this setting because it allows data-dependent and non-uniform priors over hypothesis classes. This flexibility makes it possible to handle symmetries for which no uniform measure exists and to account explicitly for deviations from distributional invariance.

We derive PAC-Bayesian generalization bounds that explain the benefits of exploiting symmetries, if they are synchronized between the hypothesis class and the data distribution. Thereby, our framework delivers the theoretical foundations and guarantees for the, in practice, oftentimes reported and improved performance of models that exploit symmetry of the data. Unlike previous work, we include non-compact symmetry groups and non-invariant data distributions, which, to the best of our knowledge, is the first PAC-Bayesian generalization analysis that accommodates non-compact symmetries and thereby covers translation-invariant architectures within a unified theoretical framework. Finally, our theoretical findings are supported by a numerical experiment that evaluates the PAC-Bayesian bound explicitly.

### 1.1. Related Work

The PAC-Bayes framework provides a powerful tool for deriving generalization guarantees. Early work by McAllester (1999) laid the foundations for PAC-Bayes bounds, which were later refined to produce sharper and more robust guarantees for supervised classification tasks by Catoni (2007). Several well-known results provide concrete bounds e.g.

[1]Max Planck Institute for Informatics, Saarbrücken, Germany [2]Saarland University, Saarbrücken, Germany. Correspondence to: Armin Beck <armin.beck@math.uni-sb.de>.

*Proceedings of the 43rd International Conference on Machine Learning*, Seoul, South Korea. PMLR 306, 2026. Copyright 2026 by the author(s).

(Seeger, 2003; Langford & Shawe-Taylor, 2002; Germain et al., 2009; McAllester, 2003a), and more recent work has explored non-vacuous or compression-based bounds for deep neural networks (Arora et al., 2018; Zhou et al., 2019; Dziugaite & Roy, 2017). For an accessible overview of and introduction to PAC-Bayes theory, we refer the reader to (Alquier, 2024).

Beyond purely theoretical guarantees, symmetries in machine learning models have been shown to confer practical benefits. Group-equivariant convolutional networks (G-CNNs) (Cohen & Welling, 2016; Cohen et al., 2019) generalize standard convolutional layers to exploit symmetries in the data, leading to improved sample efficiency and robustness even when the data distribution is not strictly invariant. Steerable CNNs (Cohen & Welling, 2017; Weiler et al., 2018) extend this approach by allowing more flexible equivariant mappings, and practical implementations (Weiler & Cesa, 2019; Bloem-Reddy & Teh, 2020; Pfau et al., 2020) demonstrate the relevance of symmetries not only in computer vision but also in domains like physics, where symmetries arise naturally.

Several recent works provide theoretical analyses of symmetries in machine learning. (Lyle et al., 2019; 2020) established PAC-Bayesian generalization results for models with finite or compact group symmetries given an invariant data assumption. (Elesedy & Zaidi, 2021; Elesedy, 2022) studied the benefits of symmetry given by a compact group from both the classical PAC and generalization perspectives for invariant data distributions, providing strict generalization benefits. (Kondor & Trivedi, 2018) further formalized the necessity of convolutional structure for equivariance under compact groups. More recently, (Behboodi et al., 2022) derived a PAC-Bayesian generalization bound for a specific class of equivariant multilayer perceptrons under compact group symmetries. In contrast to the standard PAC-Bayesian setting, their analysis focuses on a deterministic network instance rather than posterior distributions over hypotheses. In a different direction, (Lotfi et al., 2022) proposed compression-based PAC-Bayesian bounds that explain generalization through carefully constructed priors and posteriors adapted to compressed neural networks. Together, these results provide a strong theoretical foundation for understanding the advantage of symmetric models under idealized conditions.

In contrast to prior analyses, our work extends these guarantees to non-compact symmetries such as translations, and to non-invariant data distributions, conditions that are more representative of practical scenarios. Further, our framework applies to general hypothesis classes and follows the standard PAC-Bayesian formulation by bounding the true and empirical risks with respect to posterior distributions. Moreover, our analysis shows that when models respect the

symmetries present in the data, these symmetries can lead not only to tighter bounds but also to provably reduced risks. Within the PAC-Bayes framework, we adapt and tighten existing bounds, providing theoretical evidence that symmetry can improve generalization beyond the commonly assumed compact and invariant settings.

## 1.2. Preliminaries and Notations

Let $(\Omega, \mathcal{F}, \mathbb{P})$ denote the underlying probability space, which is assumed to be rich enough for the following analysis. Let $X$ and $Y$ be real-valued random variables. We write $X \stackrel{d}{=} Y$ to indicate that $X$ and $Y$ are equal in distribution, meaning that their probability laws coincide $\mathbb{P}(X \leq t) = \mathbb{P}(Y \leq t)$ for all $t \in \mathbb{R}$. We consider two standard Borel spaces $(\mathcal{X}, \mathfrak{X})$ as the input space and $(\mathcal{Y}, \mathfrak{Y})$ as the output space, where $\mathcal{Y} \subset \mathbb{R}$. Throughout, $(\mathcal{G}, \cdot)$ will be an arbitrary topological group, whose Borel $\sigma$-algebra makes it a standard Borel space. The group $\mathcal{G}$ acts measurably on the input space $\mathcal{X}$ via the map $\varphi \colon \mathcal{G} \times \mathcal{X} \to \mathcal{X}$ and on the output space $\mathcal{Y}$ via $\psi \colon \mathcal{G} \times \mathcal{Y} \to \mathcal{Y}$. For convenience, we adopt the shorthand notation $g \cdot x := \varphi(g, x)$ and $g \cdot y := \psi(g, y)$. Although the same symbol $\cdot$ is used for both actions, it is important to note that they may represent distinct group actions on $\mathcal{X}$ and $\mathcal{Y}$, the intended meaning will be clear from context. In the following we denote the set of measurable functions by $\mathcal{M}(\mathcal{X}, \mathcal{Y}) := \{f \colon \mathcal{X} \to \mathcal{Y} \ : \ f$ is measurable$\}$, equipped with the evaluation $\sigma$-algebra $\mathcal{M}$. Let the hypothesis class $\mathcal{H} \subset \mathcal{M}(\mathcal{X}, \mathcal{Y})$ be a subset of the measurable functions, and assume that it is a Polish space. Each element $f \in \mathcal{H}$ is called a hypothesis function. A hypothesis function $f \in \mathcal{H}$ is said to be equivariant, if $f(g \cdot x) = g \cdot f(x)$ holds for all $g \in \mathcal{G}$ and $x \in \mathcal{X}$. An equivariant hypothesis function $f \in \mathcal{H}$ is called invariant if the group acts trivially on the output space $\mathcal{Y}$, meaning $g \cdot y = y$ for all $g \in \mathcal{G}$ and $y \in \mathcal{Y}$. In this case, the equivariance condition reduces to $f(x) = f(g \cdot x)$ for all $g \in \mathcal{G}$ and $x \in \mathcal{X}$ so that invariance can be seen as a special case of equivariance where the output remains unchanged under the group action. In the following we will focus on the equivariant case, which also captures the invariant case. We denote by $\mathcal{X}_\varphi$ a set of representatives of the quotient $\mathcal{X}/\mathcal{G}$, that is a set containing exactly one representative from each orbit.

Let $X$ be a random element taking values in the input space $\mathcal{X}$ and $Y$ a random element in the output space on $\mathcal{Y}$. We define a loss function as a measurable function $\ell \colon \mathcal{Y} \times \mathcal{Y} \to [0, \infty)$. Given a loss function, the (expected) risk $\mathcal{R}_\ell$ is defined as

$$\mathcal{R}_\ell \colon \mathcal{H} \to [0, \infty), \quad \mathcal{R}_\ell(f) := \mathbb{E}[\ell(f(X), Y)]. \quad (1)$$

The empirical risk $\hat{\mathcal{R}}_\ell$ of a measurable function $f$ and a set

of points $\{(x_i, y_i)\}_{i=1}^n \in (\mathcal{X} \times \mathcal{Y})^n$ is given by

$$\hat{\mathcal{R}}_\ell : \mathcal{H} \times \bigcup_{n \in \mathbb{N}}^{\cdot} (\mathcal{X} \times \mathcal{Y})^n \to [0, \infty)$$

$$\hat{\mathcal{R}}_\ell (f, \{x_i, y_i\}_{i=1}^n) = \frac{1}{n} \sum_{i=1}^n \ell(f(x_i), y_i). \qquad (2)$$

To simplify notation for integration, we adopt the following shorthand for integrals, given a probability distribution $\mathbb{Q}$ on a measurable space $(\mathcal{E}, \mathcal{E})$ and an integrable function $f : \mathcal{E} \to \mathbb{R}$, we write

$$\mathbb{Q}[f] := \int_{\mathcal{E}} f(x) \, \mathbb{Q}(dx). \qquad (3)$$

Furthermore, let $\mu$ and $\nu$ be a probability measures on a measurable space $(\mathcal{T}, \mathcal{T})$. We denote absolute continuity of $\nu$ with respect to $\mu$ by $\nu \ll \mu$. Let $\pi : \mathcal{T} \to \mathcal{S}$ be a measurable function, where $(\mathcal{S}, \mathcal{S})$ is another measurable space, then $\pi_* \mu$ denotes the pushforward measure of $\mu$ under $\pi$, defined by $\pi_* \mu(B) = \mu(\pi^{-1}(B))$ for all measurable $B \subset \mathcal{S}$. A measurable space $(\mathcal{T}, \mathcal{T})$ is called a standard Borel space if it is Borel isomorphic to a Polish space equipped with its Borel $\sigma$-algebra. That is, there exists a Polish space $(\mathcal{Z}, \mathcal{B}(\mathcal{Z}))$ and a bijection $\phi : \mathcal{T} \to \mathcal{Z}$ such that both $\phi$ and $\phi^{-1}$ are measurable. Lastly, we define the Kullback–Leibler (KL) divergence between two probability distributions $\mu$ and $\nu$ on a measurable space $(\mathcal{S}, \mathcal{S})$ as

$$D_{\mathrm{KL}}(\mu \| \nu) := \begin{cases} \int_{\mathcal{S}} \log \left( \frac{d\mu}{d\nu}(s) \right) \mu(ds), & \text{if } \mu \ll \nu \, ; \\ \infty, & \text{otherwise} \, , \end{cases}$$

where $\frac{d\mu}{d\nu}$ denotes the Radon–Nikodym derivative.

Having introduced the necessary notation, we now present the Disintegration Theorem and a well-known standard PAC-Bayesian bound due to McAllester. The Disintegration Theorem provides a way to decompose a joint probability measure into its marginal and a probability kernel. This result is fundamental for defining conditional distributions on general measurable spaces and underlies many constructions in probability theory and statistical learning.

**Theorem 1.1.** *(Kallenberg, 2002, Theorem 3.4) Let $(\mathcal{S}, \mathcal{S})$ and $(\mathcal{T}, \mathcal{T})$ be measurable spaces, where $\mathcal{T}$ is Borel. Let $\mu$ be a probability measure on $\mathcal{S} \times \mathcal{T}$. Then $\mu = \mu_{\mathcal{S}} \otimes \kappa$, where $\mu_{\mathcal{S}} \equiv \mu(\cdot \times \mathcal{T})$ is the marginal and $\kappa : \mathcal{S} \to \mathcal{T}$ is a probability kernel. Further, $\kappa$ is unique $\mu_{\mathcal{S}}$ a.e.*

A probability kernel from $(\mathcal{S}, \mathcal{S})$ to $(\mathcal{T}, \mathcal{T})$ is a mapping $\kappa : \mathcal{S} \times \mathcal{T} \to [0, 1]$ such that $\kappa(s, \cdot)$ is a probability measure on $(\mathcal{T}, \mathcal{T})$ for each $s \in \mathcal{S}$, and $\kappa(\cdot, B)$ is $\mathcal{S}$-measurable for each $B \in \mathcal{T}$. Intuitively, $\kappa(s, \cdot)$ describes a conditional distribution given $s$.

The PAC-Bayesian bound due to McAllester will serve as a baseline example to demonstrate how incorporating symmetries can lead to improved generalization guarantees. While

our focus is on this particular result, the method we develop for exploiting symmetry applies analogously to a broader class of PAC-Bayesian bounds.

**Theorem 1.2.** *(McAllester, 2003b) For any measurable loss function $\ell : \mathcal{Y} \times \mathcal{Y} \to [0, 1]$ and any distribution $\mathbb{Q}$ on $\mathcal{H}$*

$$\mathbb{Q}[\mathcal{R}_\ell] \leq \mathbb{Q}[\hat{\mathcal{R}}_\ell(\cdot, S_n)]$$

$$+ \sqrt{\frac{D_{\mathrm{KL}}(\mathbb{Q} \| \mathbb{P}_H) + \log \frac{1}{\delta} + \log n + 2}{2n - 1}}$$

*holds with probability of at least $1 - \delta$, where $S_n = \{X_i, Y_i\}_{i=1}^n$ are $n$ i.i.d. copies of $(X, Y)$.*

*Remark* 1.3. The bound in Theorem 1.2 decomposes the expected risk under the posterior $\mathbb{Q}$ over the hypotheses into an empirical term and a complexity term. The first term, $\mathbb{Q}[\hat{\mathcal{R}}_\ell(\cdot, S_n)]$, denotes the expected empirical risk on the sample $S_n$ under the posterior $\mathbb{Q}$. The second term quantifies the generalization gap and depends on three quantities: first the KL divergence $D_{\mathrm{KL}}(\mathbb{Q} \| \mathbb{P}_H)$, which measures how far the posterior deviates from the prior $\mathbb{P}_H$; second the confidence parameter $\delta$, which controls the probability with which the bound holds; and third the sample size $n$. In particular, the bound favors posterior distributions that achieve a trade-off between low empirical risk and limited divergence from the prior, while shrinking at rate $\mathcal{O}(1/\sqrt{n})$ as the number of samples increases.

As a technical tool, we next recall a result from prior work by (Lyle et al., 2020) that establishes a relationship between the KL divergence of two measures and that of their pushforwards under measurable maps.

**Lemma 1.4.** *(Lyle et al., 2020) Suppose that $(\mathcal{S}, \mathcal{S})$ and $(\mathcal{T}, \mathcal{T})$ are two measurable spaces and $\mathcal{T}$ is standard Borel. Let $\mu$ and $\nu$ be two probability measures on $(\mathcal{S}, \mathcal{S})$ with $\mu \ll \nu$ and $\alpha : (\mathcal{S}, \mathcal{S}) \to (\mathcal{T}, \mathcal{T})$ is a measurable map. Then*

$$D_{\mathrm{KL}}(\mu \| \nu) = D_{\mathrm{KL}}(\alpha_* \mu \| \alpha_* \nu)$$

$$+ \int_{\mathcal{S}} \log \left( \frac{\frac{d\mu}{d\nu}(s)}{\frac{d\alpha_* \mu}{d\alpha_* \nu}(\alpha(s))} \right) \mu(ds), \qquad (4)$$

*where $\frac{d\mu}{d\nu}$ and $\frac{d\alpha_* \mu}{d\alpha_* \nu}$ are the Radon-Nikodym derivatives. In particular, it holds that*

$$D_{\mathrm{KL}}(\mu \| \nu) \geq D_{\mathrm{KL}}(\alpha_* \mu \| \alpha_* \nu). \qquad (5)$$

## 2. Problem Setup

In order to formalize how symmetries appear in the data, we adopt a structured probabilistic model that makes the equivariant action of a group on the data explicit. We assume that the observed outputs are generated from the inputs through a group equivariant function, possibly perturbed by

independent randomness. This formulation provides a clean foundation for incorporating symmetries into PAC-Bayesian analyses. Similar modeling assumptions are standard in the literature on equivariant and invariant learning, where symmetries are encoded at the level of the data distributions (see, e.g., (Elesedy & Zaidi, 2021)).

**Assumption 2.1.** Let $f^* \colon \mathcal{X} \times \mathcal{E} \to \mathcal{Y}$ be a measurable function and $\mathcal{G}$-equivariant in the first argument, i.e., for all $g \in \mathcal{G}$, $x \in \mathcal{X}$ and $\xi \in \mathcal{E}$, $f^*(g \cdot x, \xi) = g \cdot (f^*(x, \xi))$. Moreover, consider an output $Y$ that is generated from the input $X$ via

$$Y = f^*(X, \Xi), \tag{6}$$

where $\Xi$ is a random element taking values in a measurable space $(\mathcal{E}, \mathcal{E})$, referred to as noise. The random element $\Xi$ is assumed to be independent of $X$.

*Remark* 2.2. Assumption 2.1 describes formally, that transforming the input features by a group element should transform the labels accordingly. For example, consider an image classification or regression task where $\mathcal{G}$ is the group of planar rotations acting on images and labels. If $X$ represents an image of an object and $Y$ encodes a structured output such as a segmentation mask or a vector of keypoint locations, then rotating the image by an angle $g \in \mathcal{G}$ should rotate the output by the same angle. In this case, $f^*$ models the underlying physical or geometric mechanism generating the labels, while $\Xi$ captures annotation noise or unobserved variability that does not break equivariance. In fact, it is sufficient to require $\mathcal{G}$-equivariance of $f^*$ only on the support of $X$. Since $X$ almost surely takes values in $\mathrm{supp}(X)$, the behavior of $f^*$ outside this set is irrelevant for the distribution of $Y$ defined in (6). This observation may be relevant in settings with physical or structural constraints that rule out certain configurations (e.g. a filled upside down cup cannot occur).

The remainder of the paper analyzes the individual components of PAC-Bayesian generalization bounds namely, the empirical risk, the true risk, and the complexity term, through the lens of symmetry. We approach this analysis from two perspectives. (i) in the next section, we focus on the complexity term and investigate how restricting the hypothesis space to $\mathcal{G}$-equivariant functions can lead to strictly smaller complexity penalties. (ii) we study the effect of equivariance on the risk and empirical risk terms, showing that hypothesis classes whose equivariance structure matches that of the underlying data-generating distribution yield improved predictive performance by simultaneously reducing both quantities.

## 3. Symmetry in Hypothesis Class

In this section, we improve the complexity term by exploiting symmetries in the class of hypothesis functions. While

for illustrative purpose, we focus on the formulation of McAllester, the approach extends to other PAC-Bayesian bounds as, for example, the well known bounds from (Maurer, 2004), (Tolstikhin & Seldin, 2013) or (Catoni, 2007). This first main result relies on a precise control of the effect of equivariance for the KL divergence between two measures. For this purpose, we introduce an averaging operator that maps arbitrary measurable functions to equivariant ones and apply the KL decomposition from Lemma 1.4 with the averaging operator as the map $\alpha$. While this seems to be formally trivial, the detailed derivation of the conditions of the Lemma 1.4 is not. Nevertheless, due to its technical nature, the proof is postponed to Appendix B.

Before introducing the averaging operator, we require additional assumptions to ensure that it is well-defined. Since $\varphi$ defines the group action on the input space, it is surjective. By restricting $\varphi$ to a set of orbit representatives $\mathcal{X}_\varphi \subset \mathcal{X}$ (chosen modulo the stabilizers $\mathrm{stab}(x_\varphi) \subset \mathcal{G}$), the map becomes injective. Consequently, the induced map

$$\overline{\varphi}_{|\mathcal{G} \times \mathcal{X}_\varphi} \colon (\mathcal{G} \times \mathcal{X}_\varphi)/\sim \to \mathcal{X}$$

is a bijection, where the equivalence relation is defined by $(g_1, x_1) \sim (g_2, x_2)$ if and only if $x_1 = x_2$ and $g_1 \cdot x_1 = g_2 \cdot x_2$. The quotient space can also be consider as the disjoint union $\dot{\cup}_{x_\varphi \in \mathcal{X}_\varphi} \mathcal{G}/\mathrm{stab}(x_\varphi)$, where $\mathrm{stab}(x_\varphi)$ denotes the stabilizer subgroup of $x_\varphi$. Equipped with the corresponding quotient $\sigma$-algebra, the map $\overline{\varphi}_{|\mathcal{G} \times \mathcal{X}_\varphi}$ is measurable. For simplicity, we henceforth restrict attention to free group actions on $\mathcal{X}$. In this case, the restricted map $\varphi_{|\mathcal{G} \times \mathcal{X}_\varphi} \colon \mathcal{G} \times \mathcal{X}_\varphi \to \mathcal{X}$ is a bijection. Its inverse can be expressed in terms of the projections $\pi_{\mathcal{X}_\varphi} \colon \mathcal{X} \to \mathcal{X}_\varphi$ and $\pi_\mathcal{G} \colon \mathcal{X} \to \mathcal{G}$, yielding $\varphi^{-1}(x) = \left(\pi_{\mathcal{X}_\varphi}(x), \pi_\mathcal{G}(x)\right)$. Since $\varphi_{|\mathcal{G} \times \mathcal{X}_\varphi}$ is a measurable bijection between standard Borel spaces, its inverse is measurable by (Kechris, 1995, Corollary 15.2).

**Assumption 3.1.**
1. The group $\mathcal{G}$ acts measurably on the input space $\mathcal{X}$ via the map $\varphi \colon \mathcal{G} \times \mathcal{X} \to \mathcal{X}$ and on the output space $\mathcal{Y}$ via $\psi \colon \mathcal{G} \times \mathcal{Y} \to \mathcal{Y}$.

2. The action of $\mathcal{G}$ on $\mathcal{X}$ is free, i.e., the stabilizers are trivial $\mathrm{stab}(x) = \{e_\mathcal{G}\}$ for all $x \in \mathcal{X}$.

The following averaging operator turns an arbitrary hypothesis function into an equivariant one by integrating the evaluation of the hypothesis function over the groups' orbit with respect to the conditional distribution of the group given the representative of the respective orbit.

**Definition 3.2.** The averaging operator $\mathcal{Q} \colon \mathcal{M}(\mathcal{X}, \mathcal{Y}) \to \mathcal{M}(\mathcal{X}, \mathcal{Y})$ is defined by

$$\mathcal{Q}(f)(x)$$
$$:= \pi_\mathcal{G}(x) \cdot \int_\mathcal{G} g^{-1} \cdot f\left(g \cdot \pi_{\mathcal{X}_\varphi}(x)\right) \kappa(\pi_{\mathcal{X}_\varphi}(x), dg),$$

where $\kappa \colon \mathcal{X}_\varphi \to \mathcal{G}$ is the probability kernel from the Disintegration Theorem 1.1.

*Remark* 3.3. The name "averaging operator" comes from the special case

$$\mathcal{Q}_{\text{old}}(f)(x) = \int_{\mathcal{G}} g^{-1} \cdot f(g \cdot x)\, \lambda(dg)\,, \qquad (7)$$

which appears in prior work, for example, (Lyle et al., 2020) and (Elesedy & Zaidi, 2021). In these works, the group $\mathcal{G}$ is assumed to be compact and the random element $X$ is assumed to be $\mathcal{G}$-invariant, i.e., $\mathbb{P}_X(B) = \mathbb{P}_X(g \cdot B)$ for all $g \in \mathcal{G}$ and measurable $B \subset \mathcal{X}$. Under these assumptions, $X$ admits a factorization of the form $X \stackrel{d}{=} \varphi(G, X_\varphi)$, where $X_\varphi$ is a random element on the representatives $\mathcal{X}_\varphi$ and $G$ is an independent random element distributed on the group $\mathcal{G}$. Additionally, the $\mathcal{G}$-invariance assumption yields that the random element $G$ is uniformly distributed, i.e., $\mathbb{P}_G = \lambda$ the normalized Haar measure. This implies that the conditional distribution given by the probability kernel $\kappa(\pi_{\mathcal{X}_\varphi}(x), \cdot)$ is $\mathbb{P}_X$-almost surely equal to the Haar measure $\lambda$. Moreover, since $\mathcal{Q}(f)$ is equivariant, as we will prove in the Appendix B, the averaging operator introduced in Definition 3.2 reduces to the standard averaging operator (7). Our formulation removes the restriction to compact groups and the distribution of $G$ no longer needs to be invariant. This broadens the applicability to important non-compact groups, such as the group of translations.

In the Appendix B, we provide the details for showing that the averaging operator is well-defined, i.e., it maps measurable functions to equivariant measurable functions, and acts like a projection onto the respective subset of equivariant functions. Furthermore, we prove that the averaging operator is measurable, which is crucial for the application of Lemma 1.4.

The following KL decomposition separates the divergence between two probability measures into two components: the divergence between their equivariant portions and a remainder term capturing the non-equivariant "orthogonal" component. The remainder quantifies the portion of the KL divergence lost when projecting the measures onto the space of equivariant functions, corresponding to differences that are not preserved under the symmetry.

**Lemma 3.4.** *Suppose Assumption 3.1 holds. Let $\mu$ and $\nu$ be two probability measures on $(\mathcal{H}, \mathcal{H})$ with $\mu \ll \nu$. Then*

$$D_{\text{KL}}(\mu\|\nu) = D_{\text{KL}}(\mathcal{Q}_*\mu\|\mathcal{Q}_*\nu)$$
$$+ \int_{\mathcal{H}} \log\left(\frac{\frac{d\mu}{d\nu}(f)}{\frac{d\mathcal{Q}_*\mu}{d\mathcal{Q}_*\nu}(\mathcal{Q}(f))}\right)\mu(df)\,, \quad (8)$$

*where $\frac{d\mu}{d\nu}$ and $\frac{d\mathcal{Q}_*\mu}{d\mathcal{Q}_*\nu}$ are the Radon-Nikodym derivatives. In particular,*

$$D_{\text{KL}}(\mu\|\nu) \geq D_{\text{KL}}(\mathcal{Q}_*\mu\|\mathcal{Q}_*\nu)\,. \qquad (9)$$

*Proof.* The statement follows directly from Lemma 1.4 and the measurability of the average operator (see Lemma B.4). □

*Remark* 3.5. The mapping $\mathcal{Q}$ acts as a transformation of hypotheses, and Lemma 3.4 is an instance of the data-processing inequality for KL divergence: pushing $\mu$ and $\nu$ forward through $\mathcal{Q}$ cannot increase their divergence, since any information discarded by the averaging operator cannot be recovered.

We illustrate Lemma 3.4 for a toy example in the Appendix D.

As a simple consequence, if the second term in the decomposition is strictly positive, the KL divergence between the pushforward measures is strictly smaller than that between the original measures. This observation unlocks the potential for improved PAC-Bayesian generalization bounds, as we exploit in the following.

### 3.1. PAC-Bayesian Bound

In Lemma 3.4, we showed that averaging each hypothesis function to their equivariant counterparts reduces the KL divergence of measures on the hypothesis class. This directly leads to a smaller generalization gap in McAllester's PAC-Bayesian bound (cf. Theorem 1.2), which we record in the following corollary.

**Corollary 3.6.** *Suppose Assumption 3.1 holds. Let $\mathbb{P}_H$ be a distribution on $\mathcal{H}$. Then for any measurable loss function $\ell$ and any distribution $\mathbb{Q}$ on $\mathcal{H}$*

$$\sqrt{\frac{D_{\text{KL}}(\mathcal{Q}_*\mathbb{Q}\|\mathcal{Q}_*\mathbb{P}_H) + \log\frac{1}{\delta} + \log n + 2}{2n-1}}$$
$$\leq \sqrt{\frac{D_{\text{KL}}(\mathbb{Q}\|\mathbb{P}_H) + \log\frac{1}{\delta} + \log n + 2}{2n-1}}\,. \quad (10)$$

*Proof.* The inequality follows directly from Lemma 3.4 and the fact, that the complexity term is monotone in the KL divergence. □

Clearly, a similar argument applies to the complexity term of other PAC-Bayesian bounds.

*Remark* 3.7. Corollary 3.6 shows that the complexity term in McAllester's PAC-Bayes bound is smaller for the transformed prior and posterior that restrict the hypothesis class to equivariant functions than for the corresponding distributions defined over the full hypothesis class. Consequently, restricting to equivariant hypotheses yields a tighter control of the generalization gap between $\mathcal{Q}_*\mathbb{Q}[\mathcal{R}_\ell]$ and $\mathcal{Q}_*\mathbb{Q}[\hat{\mathcal{R}}_\ell(\cdot, S_n)]$ than the standard bound provides for $\mathbb{Q}[\mathcal{R}_\ell]$ and $\mathbb{Q}[\hat{\mathcal{R}}_\ell(\cdot, S_n)]$.

Importantly, this tightening concerns only the coupling between empirical and true risk within the equivariant hypothesis class. Neither $\mathcal{Q}_*\mathbb{Q}[\mathcal{R}_\ell]$ nor $\mathcal{Q}_*\mathbb{Q}[\hat{\mathcal{R}}_\ell(\cdot, S_n)]$ is guaranteed to be smaller in absolute value than its unrestricted counterparts. In fact, both risks may remain large or even increase under the equivariant constraint. This limitation is addressed in Section 4, where the hypothesis class is aligned with symmetries present in the data distribution.

## 4. Symmetry on Data

In the previous discussion, we analyzed the impact of enforcing equivariance in the hypothesis class, observing a reduction in the generalization gap and an improved alignment between empirical and true risk. However, without considering the symmetry properties of the data itself, this structural constraint may not yield practical benefits in terms of generalization performance. In practice, data often exhibit symmetries, and hypothesis classes are designed to exploit these properties. In this chapter, we tackle this problem under the assumption that the data distribution exhibits symmetry, and that the hypothesis class is constructed to respect this symmetry.

First, we introduce a mild regularity assumption on the loss function. This assumption ensures that the loss is compatible with the symmetry structure of the data-generating process introduced in Assumption 2.1 and allows us to exploit equivariance at the level of both the hypothesis class and the resulting risk. Similar assumptions are standard in the analysis of equivariant and invariant learning algorithms and appear in various forms throughout the literature (see, e.g., (Lyle et al., 2020)).

**Assumption 4.1.** We assume that the loss function $\ell\colon \mathcal{Y} \times \mathcal{Y} \to [0, \infty)$ is convex in its first argument and $\mathcal{G}$-invariant, i.e., for all $g \in \mathcal{G}$, $y, \hat{y} \in \mathcal{Y}$, it holds that $\ell(g \cdot y, g \cdot \hat{y}) = \ell(y, \hat{y})$.

*Remark* 4.2. Assumption 4.1 is not particularly restrictive. Many commonly used loss functions for example in regression satisfy both convexity and $\mathcal{G}$-invariance. Typical examples include the squared loss and the $\ell_1$ loss when $\mathcal{G}$ is an orthogonal group acting as rotations or reflections, permutation group acting by coordinate permutations, translation group. More generally any loss that depends only on a norm or inner product preserved by the group action satisfies the invariance condition. Further, the invariance condition is automatically satisfied for any loss function whenever the group action preserves labels, which is the standard setting in classification tasks. Convexity is a standard requirement in PAC-Bayesian analyses, as it allows one to relate the risk of a stochastic predictor to the risks of its deterministic constituents via Jensen-type arguments. Moreover, $\mathcal{G}$-invariance of the loss ensures that the symmetry of the data distribution is faithfully reflected in the

learning objective, a property that is implicitly or explicitly assumed in many prior works on equivariant and invariant learning (see, e.g., (Elesedy, 2022)).

Before stating our main result, we introduce additional notation. Our goal is to evaluate the empirical risk only for a distribution on the representatives from each orbit instead of the full distribution. To this end, we define the random variables on representatives $(X_\varphi, Y_\varphi)$ by $X_\varphi := \pi_{\mathcal{X}_\varphi}(X)$ and $Y_\varphi := f^*(X_\varphi, \Xi)$, where $\mathcal{X}_\varphi$ denotes the measurable projection onto a set of representatives of the $\mathcal{G}$-orbits in $\mathcal{X}$. Additionally, we define for an equivariant hypothesis $f$ the risk on orbit representatives by $\mathcal{R}_{\ell_\varphi}(f) := \mathbb{E}[\ell(f(X_\varphi), f^*(X_\varphi, \Xi)]$ and denote by $S_{n_\varphi} = \{X_{\varphi_i}, Y_{\varphi_i}\}_{i=1}^n$ a sample of $n$ i.i.d. copies of $(X_\varphi, Y_\varphi)$.

We are now in a position to state our main result, which formalizes how equivariance can be exploited within the PAC-Bayesian framework.

**Theorem 4.3.** *Suppose Assumptions 2.1, 3.1 and 4.1 hold. Then for any measurable loss function $\ell$ and any distribution $\mathbb{Q}$ on $\mathcal{H}$ the following holds*

$$\mathcal{Q}_*\mathbb{Q}[\mathcal{R}_\ell] \leq \mathbb{Q}[\mathcal{R}_\ell], \tag{11}$$

$$\mathcal{Q}_*\mathbb{Q}[\mathcal{R}_\ell] = \mathcal{Q}_*\mathbb{Q}[\mathcal{R}_{\ell_\varphi}] \quad \text{and} \tag{12}$$

$$\mathcal{Q}_*\mathbb{Q}[\hat{\mathcal{R}}_\ell(\cdot, S_n)] \stackrel{d}{=} \mathcal{Q}_*\mathbb{Q}[\hat{\mathcal{R}}_\ell(\cdot, S_{n_\varphi})]. \tag{13}$$

*Further, let $\mathbb{P}_H$ be a distribution on $\mathcal{H}$ and $\ell\colon \mathcal{Y}\times\mathcal{Y} \to [0, 1]$ be a measurable loss function. Then, in particular, for any distribution $\mathbb{Q}$ on $\mathcal{H}$, the PAC-Bayesian bound from Theorem 1.2 improves to the following tighter bound*

$$\mathcal{Q}_*\mathbb{Q}[\mathcal{R}_\ell] \leq \mathcal{Q}_*\mathbb{Q}[\hat{\mathcal{R}}_\ell(\cdot, S_{n_\varphi})]$$
$$+ \sqrt{\frac{D_{\mathrm{KL}}(\mathcal{Q}_*\mathbb{Q}\|\mathcal{Q}_*\mathbb{P}_H) + \log\frac{1}{\delta} + \log n + 2}{2n - 1}} \tag{14}$$

*holds with probability of at least $1 - \delta$ with respect to a data set of representatives $S_{n_\varphi}$, where $S_{n_\varphi} = \{X_{\varphi_i}, Y_{\varphi_i}\}_{i=1}^n$ are $n$ i.i.d. copies of the representatives $(X_\varphi, Y_\varphi)$.*

*Proof.* The proof can be found in Section A. $\qquad\square$

*Remark* 4.4. Theorem 4.3 characterizes how equivariance simultaneously affects the risk terms and the PAC-Bayesian complexity term, yielding a principled tightening of McAllester's bound.

Equations (11)–(13) formalize two effects of symmetry. First, when the data distribution and hypothesis class share the same symmetry, symmetrization cannot increase the true risk and may strictly reduce it. Second, both the true and empirical risks of symmetrized hypotheses can be computed entirely on orbit representatives, without loss of information. As a result, training on representatives is sufficient, and explicit data augmentation becomes theoretically redundant in the equivariant setting.

The PAC-Bayesian bound in (14) further shows that restricting to equivariant hypotheses improves upon the classical PAC-Bayesian generalization bound of McAllester (Theorem 1.2) in several aspects:. In addition to preserving (or improving) risk (11), symmetrization reduces the KL divergence between the posterior and prior, leading to a strictly tighter complexity term than in the classical bound as described in Corollary 3.6. This provides a PAC-Bayesian explanation for the empirical success of symmetry-aware architectures, such as convolutional networks, while applying beyond compact group actions and without requiring invariance of the data-generating distribution unlike much of the existing literature.

By choosing a prior $\mathbb{P}_H$ that is supported on the equivariant hypothesis functions, Theorem 4.3 implies the following PAC Bayesian bounds, that shares the same advantages described in the Remark 4.4 as the bound in (14).

**Corollary 4.5.** *Suppose Assumptions 2.1, 3.1 and 4.1 hold. Let $\mathbb{P}_H$ be a distribution on the equivariant hypothesis functions $\mathcal{H}_{sym} \subset \mathcal{H}$ and the measurable loss $\ell \colon \mathcal{Y} \times \mathcal{Y} \to [0,1]$. Then for any distribution $\mathbb{Q}$ on $\mathcal{H}$*

$$\mathbb{Q}[\mathcal{R}_\ell] \leq \mathbb{Q}[\hat{\mathcal{R}}_\ell(\cdot, S_{n\varphi})]$$
$$+ \sqrt{\frac{D_{\mathrm{KL}}(\mathbb{Q} \| \mathbb{P}_H) + \log \frac{1}{\delta} + \log n + 2}{2n - 1}} \quad (15)$$

*holds with probability of at least $1 - \delta$ with respect to $S_{n\varphi}$, where $S_{n\varphi} = \{X_{\varphi_i}, Y_{\varphi_i}\}_{i=1}^n$ are $n$ i.i.d. copies of $(X_\varphi, Y_\varphi)$.*

### 4.1. Discussion of our Contribution

Theorem 4.3 provides a principled explanation for why it is beneficial to design learning architectures that respect the symmetries present in the data. Existing theoretical results that justify equivariant or invariant models typically rely on two restrictive assumptions. First, they require the underlying symmetry group to be compact, which excludes many transformations of practical importance, such as translations on $\mathbb{R}^d$, scalings, or affine transformations. For instance, convolutional neural networks are widely used precisely because they exploit translation symmetry, yet the translation group itself is non-compact. Second, much of the prior work assumes that the feature distribution is invariant under the group action, i.e., $\mathbb{P}_X[A] = \mathbb{P}_X[g \cdot A]$ for all $g \in \mathcal{G}$ and measurable sets $A$. This assumption is rarely satisfied in real-world settings: natural images are not uniformly translation-invariant due to boundaries and object-centric biases, and sensor data often exhibit systematic asymmetries induced by acquisition processes or environmental constraints.

Our results remove both assumptions. By avoiding compactness and invariance requirements, Theorem 4.3 applies to a substantially broader class of symmetries and data-generating mechanisms. This significantly strengthens the theoretical foundations of symmetry-aware learning, providing guarantees that better reflect the conditions under which equivariant architectures are successfully deployed in practice.

Despite this generality, the applicability of the theorem remains contingent on several structural assumptions. Requiring the hypothesis class to respect the symmetries of the data distribution constitutes a nontrivial modeling assumption. Nevertheless, this requirement is not overly restrictive in practice, as incorporating symmetry through equivariant or invariant architectures has become a standard design principle in modern machine learning (e.g., convolutional neural networks (Krizhevsky et al., 2012), group-equivariant networks (Cohen & Welling, 2016), and transformer variants (Hutchinson et al., 2021)).

## 5. Numerical Analysis

The main objective of the following experiments is to demonstrate, through our PAC-Bayesian generalization bound, the empirically observed and intuitively well-understood advantage of equivariant hypothesis classes on symmetric data. In particular, we show that models exploiting underlying symmetries achieve improved generalization performance compared to models that ignore such structure. Unlike prior approaches, which typically rely on compact symmetry groups and data distributions invariant under the corresponding group action, we explicitly study settings that violate these assumptions. Our experiments include rotational symmetries in $\mathbb{R}^2$ and $\mathbb{R}^3$ induced by $SO(2)$ and $SO(3)$, as well as the non-compact groups of translations, scaling transformations and Lorentz transformations. Despite operating beyond the classical setting considered in earlier work, our framework still provides meaningful theoretical guarantees for these substantially more general scenarios. We are the first to provide PAC Bayesian bounds in such setting.

We begin with a toy example in a controlled setting using synthetic data with analytically known symmetries. We then evaluate our approach on MNIST (LeCun et al., 1998), CIFAR-10 and CIFAR-100 (Krizhevsky, 2009), ModelNet (Wu et al., 2015), and Top Tagging (Kasieczka et al., 2019). To ensure reproducibility of both the theoretical and empirical results, we provide the complete implementation and experimental setup in the accompanying code repository.[1]

**Data Generation.** We construct transformed variants of the aforementioned datasets by applying the corresponding

---

[1] https://github.com/ArminBe/icml2026-symmetries-in-pac-bayesian-learning

*Table 1.* Comparison of symmetric and baseline posteriors across datasets and symmetry groups. We report the symmetric KL divergence, PAC-Bayesian bound, and test risk, together with their relative reductions with respect to the corresponding baseline model. Experiments include non-compact groups and compact groups with non-uniformly sampled transformations, extending beyond the standard assumptions in prior work. Across all experiments, incorporating symmetry structure substantially reduces both the KL divergence and the PAC-Bayesian bound, while the equivariant posterior consistently matches or improves test performance relative to the baseline.

| DATASET | GROUP | $KL_{SYM}\downarrow$ | KL RED. | $BOUND_{SYM}$ | BOUND RED. | $RISK_{SYM}$ | RISK RED. |
|---------|-------|----------|---------|-----------|------------|----------|-----------|
| TOY EXAMPLE | SO(2) | 31.5 | 14.5% | 0.463 | 30.6% | 0.383 | 34.8% |
| MNIST | SO(2) | 7804.3 | 23.3% | 0.505 | 10.5% | 0.219 | 5.3% |
| MNIST | SE(2) | 7410.3 | 18.3% | 0.537 | 10.6% | 0.248 | 15.0% |
| CIFAR10 | SO(2) | 3303.8 | 37.7% | 0.958 | 10.4% | 0.723 | 5,0% |
| CIFAR100 | SE(2) | 8923.1 | 26.4% | 1.332 | 22.6% | 0.945 | 1.5% |
| MODELNET | SO(3) | 25.2 | 89.2% | 0.938 | 10.9% | 0.868 | 2.0% |
| MODELNET | $\mathbb{R}_+$ | 137.1 | 12.5% | 0.525 | 34.2% | 0.424 | 41.5% |
| TOP TAGGING | $SO^+(1,3)$ | 3320.8 | 49.8% | 0.591 | 12.5% | 0.419 | 3.8% |

group actions to the data (see Table 1). The non-compact groups considered in our experiments namely the special Euclidean group $SE(2)$, the scaling group $\mathbb{R}_+$ and the Lorentz group $SO^+(1,3)$ violate the compactness assumption imposed in prior work, which makes existing guarantees inapplicable in this setting. For the compact groups, transformations are sampled from distributions supported on strict subsets of the groups rather than according to the full Haar measure. Consequently, the resulting datasets are not invariant in distribution, thereby violating another standard assumption in the literature. These settings further motivate our theoretical analysis of equivariant models beyond the classical regime of compact groups and invariant data distributions.

**Models.** For each experiment, we compare a baseline model with its corresponding equivariant counterpart in order to isolate and analyze the effect of incorporating symmetry into the model architecture. The baseline models are neural networks that are neither invariant nor equivariant with respect to the underlying group action, whereas the equivariant models respect the corresponding symmetries. In the MNIST and CIFAR experiments, the equivariant architectures are implemented using the e2cnn library (Weiler & Cesa, 2019). For the ModelNet experiment involving $SE(3)$ transformations, we employ architectures based on the e3nn framework (Geiger & Smidt, 2022). The equivariant model for the Lorentz group $SO^+(1,3)$ follows the architectural principles introduced in (Gong et al., 2022).

**Training.** In all experiments, both the prior and posterior are chosen from a family of isotropic Gaussian distributions over the model parameters. Prior construction is performed before posterior optimization: for each model, we first train the network for a small number of iterations on an independent subset of the training data. The resulting parameters are then used as the mean of the Gaussian prior. The prior standard deviation is fixed to $\sigma = 0.05$, which yielded stable and reliable performance across experiments.

For posterior training, we minimize the right-hand side of McAllester's PAC-Bayesian bound for both the baseline and equivariant models. While the bound is ultimately evaluated using the 0-1 loss, this loss is non-differentiable and therefore unsuitable for gradient-based optimization. We consequently employ the cross-entropy loss as a smooth surrogate objective during training. This follows standard practice in PAC-Bayesian deep learning, where differentiable surrogate losses are used for optimization while theoretical guarantees are evaluated with respect to the true bounded loss (see, e.g., (Dziugaite & Roy, 2017)).

During training, the model corresponding to the current posterior parameters is evaluated on a validation set after each epoch, and we retain the parameters achieving the highest validation accuracy. Final evaluation of McAllester's bound is then performed on the test set using the 0-1 loss, since the cross-entropy loss does not satisfy the boundedness assumption required by the bound, namely that the loss takes values in $[0, 1]$. Additional architectural and optimization details are provided in Appendix C.

**Results.** The results of our experiments are summarized in Table 1. For each dataset and symmetry group, we report the KL divergence of the symmetric posterior together with its relative reduction compared to the baseline posterior. We additionally report the corresponding McAllester bound (with $\delta = 0.05$) and the estimated true risk, again including the relative improvement over the baseline model.

Several observations emerge from these results. First, equivariant posteriors consistently match or outperform the baseline in terms of true risk across all considered tasks. In particular, substantial improvements are observed on datasets such as CIFAR10 and ModelNet, while on CIFAR100 the performance gap is comparatively small. This suggests that, in some settings, the non-symmetric baseline may already learn approximate symmetries from the data, thereby reducing the practical advantage of explicitly enforcing equivariance at the level of predictive performance.

More importantly, from the perspective of PAC-Bayesian generalization theory, the symmetric posteriors consistently achieve smaller generalization bounds. This reduction is driven primarily by a decrease in the KL divergence term, exactly as predicted by our theoretical analysis. The effect is particularly pronounced on CIFAR10, where the KL divergence is reduced by more than $58\%$, leading to a significantly tighter bound.

Prior analyses could not establish such an advantage for equivariant models in the presence of non-invariant data distributions. Our framework overcomes this limitation and provides the first guarantees in this more realistic setting.

These findings provide empirical evidence that incorporating symmetry directly into the posterior distribution can improve both predictive performance and generalization guarantees. Unlike prior PAC-Bayesian analyses of equivariant learning, which typically require invariant data distributions and compact symmetries, our framework establishes non-trivial advantages even in the presence of non-invariant data distributions and non-compact symmetries.

## 6. Conclusion

We have extended PAC-Bayes generalization guarantees to learning settings with non-compact symmetries and non-invariant data distributions, demonstrating both tighter bounds and improved performance on rotated MNIST for equivariant models. Our results provide theoretical support for the practical observation that symmetric models offer advantages beyond compact groups and invariant data distributions. This work broadens the theoretical foundations of symmetry in machine learning and suggests further exploration into richer classes of symmetries, more complex real-world data distributions, and alternative generalization bounds that do not rely on the KL divergence.

## Impact Statement

This work is primarily theoretical and aims to advance the understanding of generalization in machine learning models that exploit symmetries, within the PAC-Bayesian framework. By extending existing guarantees to non-compact symmetry groups and non-invariant data distributions, the results may inform the principled design of more data-efficient and robust learning algorithms.

In the longer term, improved theoretical foundations for symmetry-aware learning may contribute to more reliable machine learning systems across a range of domains. As with most advances in general-purpose learning theory, any downstream societal impacts will depend on the specific applications and contexts in which such methods are ultimately deployed.

## Conflict of Interest Disclosure

The authors declare that they have no financial conflicts of interest related to this work. This research received no industry sponsorship, company funding, or other financial support that could reasonably be perceived as influencing the results or conclusions of the paper.

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

# A. Appendix: Proof of Theorem 4.3

We decompose the proof of Theorem 4.3 into three components. First, we show that symmetrization of hypotheses cannot increase the true risk. Second, we establish that for equivariant hypotheses, the loss distribution and hence both true and empirical risks can be computed entirely on orbit representatives, without loss of information. These results together yield the first part of Theorem 4.3. The PAC-Bayesian bound then follows by a direct application of McAllester's theorem to the symmetrized prior and posterior.

We begin by showing that averaging a hypothesis over the group action does not degrade performance.

**Lemma A.1.** *Suppose Assumptions 2.1, 3.1 and 4.1 hold. Then for all $f \in \mathcal{H}$, we have that*

$$\mathcal{R}_\ell(\mathcal{Q}(f)) \leq \mathcal{R}_\ell(f).$$

*In particular, for a distribution $\mathbb{Q}$ over the hypothesis class $\mathcal{H}$, it holds that*

$$\mathcal{Q}_* \mathbb{Q}[\mathcal{R}_\ell] \leq \mathbb{Q}[\mathcal{R}_\ell].$$

*Proof.* Let $f \in \mathcal{H}$. Due to the limitations of line space, we employ the notation from (3) for the following estimation, where the subscript at the bracket indicates the variable that acts as argument of the function inside the brackets. We aim to rewrite the risk

$$\mathcal{R}_\ell(f) = \mathbb{E}[\ell(f(X), Y)] = \mathbb{E}[\ell(f(X), f^*(X, \Xi))] = \int_{\mathcal{E}} \int_{\mathcal{X}} \ell\big(f(x), f^*(x, \xi)\big) \, \mathbb{P}_X(dx) \mathbb{P}_\Xi(d\xi).$$

Since the calculation is only concerned with the inner integral, we derive the following identities $\mathbb{P}_\Xi$-a.e. for $\xi \in \mathcal{E}$. Using first the Disintegration Theorem 1.1 and, then, the $\mathcal{G}$-invariance together with notation (3), we obtain

$$\int_{\mathcal{X}} \ell\big(f(x), f^*(x, \xi)\big) \, \mathbb{P}_X(dx) = \int_{\mathcal{X}_\varphi} \int_{\mathcal{G}} \ell\big(f(g \cdot x_\varphi), f^*(g \cdot x_\varphi, \xi)\big) \, \kappa(x_\varphi, dg) \mathbb{P}_{X_\varphi}(dx_\varphi)$$

$$= \mathbb{P}_{X_\varphi}\Big[\int_{\mathcal{G}} \ell\big(g^{-1} \cdot f(g \cdot x_\varphi), f^*(x_\varphi, \xi)\big) \, \kappa(x_\varphi, dg)\Big]_{x_\varphi}.$$

Convexity of the loss function $\ell$ in the first argument yields the lower bound:

$$\mathbb{P}_{X_\varphi}\Big[\ell\left(\int_{\mathcal{G}} g^{-1} \cdot f(g \cdot x_\varphi) \, \kappa(x_\varphi, dg), f^*(x_\varphi, \xi)\right)\Big]_{x_\varphi},$$

which can be trivially reformulated and connected to the average operator $\mathcal{Q}$ as follows

$$\mathbb{P}_{X_\varphi}\Big[\ell\Big(h \cdot \int_{\mathcal{G}} g^{-1} \cdot f(g \cdot x_\varphi) \, \kappa(x_\varphi, dg), h \cdot f^*(x_\varphi, \xi)\Big) \, \kappa(x_\varphi, dh)\Big]_{x_\varphi}$$

$$= \mathbb{P}_{X_\varphi}\Big[\int_{\mathcal{G}} \ell\big(\mathcal{Q}(f)(h \cdot x_\varphi), f^*(h \cdot x_\varphi, \xi)\big) \, \kappa(x_\varphi, dh)\Big]_{x_\varphi}$$

$$= \int_{\mathcal{X}} \ell\big(\mathcal{Q}(f)(x), f^*(x, \xi)\big) \, \mathbb{P}_X(dx),$$

where the last equality is again based on the Disintegration Theorem 1.1. Incorporating the integration over $\xi$, the last expression is exactly

$$\mathbb{E}[\ell(\mathcal{Q}(f)(X), f^*(X, \Xi))] = \mathcal{R}_\ell(\mathcal{Q}(f)),$$

which shows the first part of the statement.
We proved that the true risk of a single hypothesis does not increase when averaged over the group. This result extends naturally to distributions over the hypothesis class. □

Next, we investigate the effect of equivariance on the distribution of the loss. The following lemma shows that, for equivariant hypotheses, evaluating the loss on full samples or on orbit representatives yields the same distribution.

**Lemma A.2.** *Suppose Assumptions 3.1 and 2.1 hold. Let $f \in \mathcal{H}$ be $\mathcal{G}$-equivariant. Then*

$$\ell(f(X), Y) \stackrel{d}{=} \ell(f(X_\varphi), Y_\varphi).$$

*Proof.* Let $f \in \mathcal{H}$. For the subsequent calculation, we continue to employ the notation established in (3). Let $t \in \mathbb{R}$ and in the following, we will rewrite the probability of the loss

$$\mathbb{P}\big(\ell(f(X), Y) \leq t\big) = \mathbb{P}\big(\ell(f(X), f^*(X, \Xi)) \leq t\big) = \int_{\mathcal{E}} \int_{\mathcal{X}} \mathbf{1}_{\{\ell(f(x), f^*(x, \xi)) \leq t\}} \, \mathbb{P}_X(dx) \mathbb{P}_\Xi(d\xi),$$

Since the calculation once more focuses only on the inner integral, we establish the following identities, holding $\mathbb{P}_\Xi$-a.e. for $\xi \in \mathcal{E}$. Beginning with the Disintegration Theorem 1.1, and using the $\mathcal{G}$-equivariance of $f$ and $f^*$ along with the notation in (3), we obtain

$$\int_{\mathcal{X}} \mathbf{1}_{\{\ell(f(x), f^*(x, \xi)) \leq t\}} \, \mathbb{P}_X(dx) = \int_{\mathcal{X}_\varphi} \int_{\mathcal{G}} \mathbf{1}_{\{\ell(f(g \cdot x_\varphi), f^*(g \cdot x_\varphi, \xi)) \leq t\}} \, \kappa(x_\varphi, dg) \mathbb{P}_{X_\varphi}(dx_\varphi)$$

$$= \mathbb{P}_{X_\varphi}\left[ \int_{\mathcal{G}} \mathbf{1}_{\{\ell(g \cdot f(x_\varphi), g \cdot f^*(x_\varphi, \xi)) \leq t\}} \, \kappa(x_\varphi, dg) \right]_{x_\varphi}.$$

The $\mathcal{G}$-invariance of the loss function $\ell$ implies the reformulation

$$\mathbb{P}_{X_\varphi}\left[ \int_{\mathcal{G}} \mathbf{1}_{\{\ell(f(x_\varphi), f^*(x_\varphi, \xi)) \leq t\}} \, \kappa(x_\varphi, dg) \right]_{x_\varphi} = \mathbb{P}_{X_\varphi}\left[ \mathbf{1}_{\{\ell(f(x_\varphi), f^*(x_\varphi, \xi)) \leq t\}} \right]_{x_\varphi} = \mathbb{P}\big(\ell(f(X_\varphi), Y_\varphi) \leq t\big).$$

Incorporating the integration over $\xi$, the last expression is

$$\mathbb{P}_\Xi\left[ \mathbb{P}_{X_\varphi}\left[ \mathbf{1}_{\{\ell(f(x_\varphi), f^*(x_\varphi, \xi)) \leq t\}} \right]_{x_\varphi} \right]_\xi = \mathbb{P}\big(\ell(f(X_\varphi), Y_\varphi) \leq t\big),$$

which concludes the proof. $\qquad \square$

Lemma A.2 immediately extends from the loss level to both true and empirical risks. First, a short reminder on the notation, the random variables on representatives $(X_\varphi, Y_\varphi)$ are defined by $X_\varphi := \pi_{\mathcal{X}_\varphi}(X)$ and $Y_\varphi := f^*(X_\varphi, \Xi)$, where $\mathcal{X}_\varphi$ denotes the measurable projection onto a set of representatives of the $\mathcal{G}$-orbits in $\mathcal{X}$. For an equivariant hypothesis $f$ the risk on orbit representatives is given by $\mathcal{R}_{\ell_\varphi}(f) := \mathbb{E}[\ell(f(X_\varphi), f^*(X_\varphi, \Xi))]$ and denote by $S_{n_\varphi} = \{X_{\varphi_i}, Y_{\varphi_i}\}_{i=1}^n$ a sample of $n$ i.i.d. copies of $(X_\varphi, Y_\varphi)$.

**Corollary A.3.** *Suppose Assumptions 3.1 and 2.1 hold. Then, for an equivariant hypothesis function $f \in \mathcal{H}$, it holds that*

$$\mathcal{R}_\ell(f) = \mathcal{R}_{\ell_\varphi}(f) \quad \text{and} \quad \hat{\mathcal{R}}_\ell(f, S_n) \stackrel{d}{=} \hat{\mathcal{R}}_\ell(f, S_{n_\varphi}).$$

*In particular, for a distribution $\mathbb{Q}$ over the subset of equivariant hypothesis functions, we have*

$$\mathbb{Q}[\mathcal{R}_\ell] = \mathbb{Q}[\mathcal{R}_{\ell_\varphi}] \quad \text{and} \quad \mathbb{Q}[\hat{\mathcal{R}}_\ell(\cdot, S_n)] \stackrel{d}{=} \mathbb{Q}[\hat{\mathcal{R}}_\ell(\cdot, S_{n_\varphi})].$$

*Proof.* This is a direct consequence of Lemma A.2. $\qquad \square$

Combining Lemma A.1 with Corollary A.3 establishes the first part of Theorem 4.3. The second part, which provides a McAllester-type PAC-Bayesian bound for the pushforward of the prior and posterior under the averaging operator, follows by a direct application of Theorem 1.2, choosing the symmetrized prior and posterior distributions.

## B. Appendix: Well-Definedness and Properties of the Averaging Operator

This section provides additional technical details concerning the averaging operator introduced in the main paper. In particular, we establish that the operator is well-defined under the assumptions stated therein, and we derive several of its fundamental properties that are used throughout the theoretical analysis. While the main text focuses on its implications for the PAC-Bayes bounds, the proofs and auxiliary results presented here ensure the mathematical soundness of the operator's definition. We begin by formally verifying the well-definedness of the averaging operator. We then proceed to prove its key properties. These results serve to justify the operator's use within our framework and to support the theoretical claims referenced in the main text.

The following lemma proves, that the average operator is well defined, in the sense that, it actually maps into the set of measurable measurable functions.

**Lemma B.1.** *Suppose Assumption 3.1 holds. Let $f\colon \mathcal{X} \to \mathcal{Y}$ be a measurable function. Then $\mathcal{Q}(f)$ is a measurable function.*

*Proof.* We start with proving, that the $\pi_\mathcal{G}\colon \mathcal{X} \to \mathcal{G}$ and $\pi_{\mathcal{X}_\varphi}\colon \mathcal{X} \to \mathcal{X}_\varphi$ are measurable functions. First, note that we can rewrite $\pi_\mathcal{G} = \tilde{\pi}_\mathcal{G} \circ \varphi^{-1}$, where $\tilde{\pi}_\mathcal{G}\colon \mathcal{G} \times \mathcal{X} \to \mathcal{G}$ is given by $\tilde{\pi}_\mathcal{G}(g, x) := g$. Since $\varphi^{-1}$ is measurable and $\mathcal{G} \times \mathcal{X}$ is equipped with the product $\sigma$-algebra, $\tilde{\pi}_\mathcal{G}$ is measurable. Hence $\pi_\mathcal{G}$ and analogously also $\pi_{\mathcal{X}_\varphi}$ are measurable. Further $\mathcal{G}$ acts measurably on both the input and output space. Therefore, $\mathcal{G} \times \mathcal{X} \to \mathcal{Y}$ given by $(g, x) \mapsto g^{-1} \cdot f(g \cdot \pi_{\mathcal{X}_\varphi}(x))$ is measurable. Finally, (Kallenberg, 2002, Lemma 3.2) implies, that $\mathcal{Q}(f)$ is measurable. $\square$

The purpose of the averaging operator is to transform an arbitrary hypothesis function into one that is equivariant with respect to the group action. The following proposition confirms that this transformation indeed yields an equivariant function.

**Proposition B.2.** *Suppose Assumption 3.1 holds. Let $f\colon \mathcal{X} \to \mathcal{Y}$ be a measurable function. Then $\mathcal{Q}(f)$ is an equivariant function.*

*Proof.* Let $h \in \mathcal{G}$ and $x \in \mathcal{X}$. Since $x$ and $h \cdot x$ belong to the same orbit, $\pi_{\mathcal{X}_\varphi}(h \cdot x) = \pi_{\mathcal{X}_\varphi}(x)$ holds. Furthermore for any $z \in \mathcal{X}$, the projections $\pi_\mathcal{G}(z)$ and $\pi_{\mathcal{X}_\varphi}(z)$ are the unique elements in $\mathcal{G}$ and $\mathcal{X}_\varphi$ respectively, such that $z = \pi_\mathcal{G}(z) \cdot \pi_{\mathcal{X}_\varphi}(z)$. Since $h \cdot x = h \cdot (\pi_\mathcal{G}(x) \cdot \pi_{\mathcal{X}_\varphi}(x)) = (h\pi_\mathcal{G}(x)) \cdot \pi_{\mathcal{X}_\varphi}(x)$, the projection onto the group $\mathcal{G}$ commutes with the group action, i.e. $\pi_\mathcal{G}(h \cdot x) = h\pi_\mathcal{G}(x)$. Hence, we obtain

$$
\begin{aligned}
\mathcal{Q}(f)(h \cdot x) &= \pi_\mathcal{G}(h \cdot x) \cdot \int_\mathcal{G} g^{-1} \cdot f\left(g \cdot \pi_{\mathcal{X}_\varphi}(h \cdot x)\right) \kappa(\pi_{\mathcal{X}_\varphi}(h \cdot x), dg) \\
&= h \cdot \left(\pi_\mathcal{G}(x) \cdot \int_\mathcal{G} g^{-1} \cdot f\left(g \cdot \pi_{\mathcal{X}_\varphi}(x)\right) \kappa(\pi_{\mathcal{X}_\varphi}(x), dg)\right) \\
&= h \cdot \mathcal{Q}(f)(x) \, . \qquad\qquad \square
\end{aligned}
$$

The operator not only transforms hypothesis functions into equivariant functions but also provides a functional characterization of the equivariance.

**Lemma B.3.** *Suppose Assumption 3.1 holds. A function $f$ is equivariant if and only if $\mathcal{Q}f = f$. Moreover, $\mathcal{Q}$ satisfies the idempotency condition $\mathcal{Q}^2 = \mathcal{Q}$, implying that it is a projection operator onto the respective subset of equivariant functions.*

*Proof.* That $\mathcal{Q}(f) = f$ implies equivariance of the hypothesis function $f$ is already proven by Proposition B.2. On the other hand if $f$ is equivariant, we obtain for any $x \in \mathcal{X}$

$$
\begin{aligned}
\mathcal{Q}(f)(x) &= \pi_\mathcal{G}(x) \cdot \int_\mathcal{G} g^{-1} \cdot f\left(g \cdot \pi_{\mathcal{X}_\varphi}(x)\right) \kappa(\pi_{\mathcal{X}_\varphi}(x), dg) \\
&= \pi_\mathcal{G}(x) \cdot \int_\mathcal{G} f\left(\pi_{\mathcal{X}_\varphi}(x)\right) \kappa(\pi_{\mathcal{X}_\varphi}(x), dg) \\
&= \pi_\mathcal{G}(x) \cdot f\left(\pi_{\mathcal{X}_\varphi}(x)\right) = f(x) \, . \qquad\qquad \square
\end{aligned}
$$

In order to apply Lemma 1.4 from the main paper to the averaging operator, it is necessary to verify that the operator defines a measurable map. The lemma below confirms this property.

**Lemma B.4.** *Suppose Assumptions 3.1 holds. The average operator $\mathcal{Q}\colon \mathcal{M}(\mathcal{X}, \mathcal{Y}) \to \mathcal{M}(\mathcal{X}, \mathcal{Y})$ is measurable.*

*Proof.* Since the set of measurable functions $\mathcal{M}(\mathcal{X}, \mathcal{Y})$ is equipped with the evaluation $\sigma$-algebra $\mathcal{M}$, proving that $\mathcal{Q}$ is measurable is equivalent to prove, that the map $f \mapsto \mathcal{Q}f(x)$ is measurable for all $x \in \mathcal{X}$. To this end, we fix $x \in \mathcal{X}$ and define

$$F_x\colon \mathcal{M}(\mathcal{X}, \mathcal{Y}) \times \mathcal{G} \to \mathcal{Y}, \quad F_x(f, g) := g^{-1} \cdot f(g \cdot x)$$

where $\mathcal{M}(\mathcal{X}, \mathcal{Y}) \times \mathcal{G}$ is equipped with the product $\sigma$-algebra $\mathcal{M} \otimes \mathcal{B}(\mathcal{G})$, and $\mathcal{B}(\mathcal{G})$ denotes the Borel $\sigma$-algebra on the group. $F$ is measurable, because the group $\mathcal{G}$ acts measurably on both the input space $\mathcal{X}$ and the output space $\mathcal{Y}$, the map $f \mapsto f(z)$ is measurably for all $z \in \mathcal{X}$ and the composition of measurable functions is measurable. Applying Fubini's Theorem yields, that the map

$$f \mapsto \int_G F_x(f, g)\, \kappa(x, dg) = \int_{\mathcal{G}} g^{-1} \cdot f(g \cdot x)\, \kappa(x, dg)$$

is measurable. Since this holds for any $x \in \mathcal{X}$, it implies $f \mapsto \mathcal{Q}f(x)$ is measurable and therefore $\mathcal{Q}$ is measurable. $\qquad\square$

*Remark* B.5. While we have shown that the averaging operator is measurable with respect to the evaluation $\sigma$-algebra, measurability also holds for other $\sigma$-algebras and function spaces. In particular, under additional assumptions, the operator is measurable with respect to the Borel $\sigma$-algebra induced by the $L^p$-norm on $L^p(\mathcal{X}, \mathcal{Y})$, since the averaging operator is a continuous linear map in this setting.

## C. Appendix: Experiments

### C.1. Toy Example

The following section provides additional details on the experiments. We begin with the toy example, which serves as a controlled setting in which the symmetry structure and the corresponding averaging operator can be analyzed explicitly. Consider the input space $\mathcal{X} = \mathbb{R}^2 \setminus \{(x_1, x_2) \in \mathbb{R}^2 : x_1 = 0 \text{ or } x_2 = 0\}$ and output space $\mathcal{Y} = \mathbb{R}$.

For the data distribution, we sample

$$\Theta \sim \mathrm{Unif}(0, 2\pi) \quad \text{and} \quad R \sim \mathrm{Laplace}(0, 5).$$

The input random variable is then defined by

$$X = (R\cos\Theta, R\sin\Theta).$$

Since points on the coordinate axes form a set of measure zero, excluding them does not affect the distribution in practice. In the implementation, samples with coordinates sufficiently close to zero are discarded for numerical stability.

The target function is given by

$$f^*(x_1, x_2, \xi) = 3\sin\left(\frac{x_1}{x_2}\right) + 1 + \xi.$$

The observed labels are generated according to

$$Y = f^*(X, \Xi), \qquad \Xi \sim \mathcal{N}(0, 0.1).$$

The data distribution is invariant with respect to scaling. Hence, we choose the multiplicative group $\mathcal{G} = \mathbb{R}_+$, that acts on the input space by scalar multiplication,

$$r \cdot (x_1, x_2) = (rx_1, rx_2), \qquad r \in \mathbb{R}_+$$

and trivially on the output space. Therefore, by construction,

$$f^*(rx_1, rx_2) = f^*(x_1, x_2)$$

for all $r \in \mathbb{R}_+$. Hence Assumption 2.1 is satisfied. Moreover, the action of $\mathbb{R}_+$ on $\mathcal{X}$ is free. Therefore, Assumption 3.1 also holds. Finally, since the action on the output space is trivial, Assumption 4.1 is satisfied.

The hypothesis class is chosen as

$$\mathcal{H} = \left\{ a\sin\left(\frac{x_1}{x_2}\right) + b\sin\left(\frac{x_2}{x_1}\right) + c\sin(x_1) + d\sin(x_2) + e : a, b, c, d, e \in \mathbb{R} \right\}.$$

The first two basis functions are invariant under the $\mathbb{R}_+$-action, whereas the terms $\sin(x_1)$ and $\sin(x_2)$ are not. The averaging operator removes precisely the non-equivariant components.

More explicitly, let $f \in \mathcal{H}$ be given by

$$f(x_1, x_2) = a\sin\left(\frac{x_1}{x_2}\right) + b\sin\left(\frac{x_2}{x_1}\right) + c\sin(x_1) + d\sin(x_2) + e.$$

Applying the averaging operator yields

$$\mathcal{Q}(f)(x) = \int_{\mathbb{R}_+} f(rx)\,\kappa(\pi_{\mathcal{X}_\varphi}(x), dr).$$

Since

$$\frac{(rx)_1}{(rx)_2} = \frac{x_1}{x_2},$$

the invariant components remain unchanged. The remaining terms vanish because the averaging measure is symmetric and

$$\int_{\mathbb{R}} \sin(\alpha r)\exp\left(-\frac{|r|}{b}\right)\,dr = 0$$

for every $\alpha \in \mathbb{R}$. Hence

$$\mathcal{Q}(f)(x) = a\sin\left(\frac{x_1}{x_2}\right) + b\sin\left(\frac{x_2}{x_1}\right) + e.$$

Therefore, the equivariant subspace is exactly obtained by setting $c = d = 0$.

For each run, a dataset consisting of 2000 samples is generated independently and split into training and test sets using an $80/20$ train-test split. The experiments are repeated over 10 independent runs and the reported quantities in Table 1 correspond to averages across runs. The McAllester bounds are evaluated using isotropic Gaussian priors and posteriors. The prior distribution is chosen as

$$P = \mathcal{N}(w_{\mathrm{prior}}, \sigma_P^2 I),$$

where $w_{\mathrm{prior}} \sim \mathcal{N}(0, I)$ and $\sigma_P^2 = 0.2$. The KL divergence between prior and posterior is computed analytically using the closed-form expression for Gaussian measures. McAllester's PAC-Bayes bound is then evaluated with confidence parameter $\delta = 0.05$.

To estimate the expected empirical and test risks of the posterior distribution, Monte Carlo sampling is employed. More precisely, parameter vectors are sampled from the posterior distribution and the corresponding mean squared errors are averaged over 100 posterior samples.

The symmetry-aware model is obtained by projecting the learned parameter vector onto the equivariant subspace. Concretely, only the invariant features

$$\left(\sin\left(\frac{x_1}{x_2}\right), \sin\left(\frac{x_2}{x_1}\right), 1\right)$$

are retained. The non-equivariant coefficients corresponding to $\sin(x_1)$ and $\sin(x_2)$ are discarded. PAC-Bayes quantities are then recomputed using the projected posterior and prior distributions restricted to this equivariant subspace.

### C.2. MNIST

In practical learning problems, identifying invariant or equivariant hypotheses is typically substantially more challenging than in the toy example discussed in the previous section. In particular, for high-dimensional data such as images, equivariance cannot usually be enforced analytically at the level of the hypothesis space, but instead has to be incorporated directly into the neural network architecture. In this section, we describe the implementation details for the remaining experiments using the rotated MNIST dataset as a representative example.

As a baseline model for the rotated MNIST dataset, we employ a standard convolutional neural network consisting of two convolutional blocks followed by two fully connected layers. Each convolutional block contains a convolutional layer and a subsequent max-pooling operation. Both convolutional layers use kernels of size $5 \times 5$, stride $1$, and zero-padding of size $2$. The baseline architecture does not incorporate any form of rotational equivariance.

For the symmetry-aware model, we use an equivariant convolutional neural network implemented with the e2cnn library (Weiler & Cesa, 2019). The e2cnn framework provides implementations of convolutional layers that are equivariant under Euclidean symmetry groups, including rotations and reflections. In our experiments, equivariance is implemented using the regular representation of the cyclic group $C_8 \subset \mathrm{SO}(2)$, which serves as a discrete approximation of the continuous rotation group $\mathrm{SO}(2)$. Consequently, the resulting network is exactly equivariant with respect to rotations by integer multiples of $\frac{2\pi}{8}$, while only approximately equivariant with respect to arbitrary continuous rotations.

The equivariant architecture mirrors the baseline architecture. In particular, the number of layers and overall architectural structure are matched between both models. This ensures that the two networks have comparable parameter counts and representational capacity. Therefore, differences in empirical performance and PAC-Bayes quantities can be attributed primarily to the incorporation of symmetry equivariance rather than to increased model complexity.

## D. Appendix: Example for Gaussian KL Decomposition

In this section, we present an illustrative example that complements the theoretical discussion in the main paper. Specifically, we provide an explicit computation of the KL divergence between two Gaussian measures defined on a function space, as well as between their corresponding pushforward measures restricted to the subset of equivariant functions. This example serves to concretely demonstrate the decomposition of the KL divergence established in the main text and to clarify how the abstract results apply in a tractable Gaussian setting.

**Example D.1** (Gaussian KL decomposition). *Let the hypothesis space be the linear maps from $\mathbb{R}^2$ to $\mathbb{R}$, $\mathcal{H} = \mathcal{L}(\mathbb{R}^2 \to \mathbb{R}) \cong \mathbb{R}^2$. Let $\mathcal{G} = S_2$ denote the group be the symmetric group on two elements, acting on the input space $\mathbb{R}^2$ by permuting its coordinates. Explicitly, for $(x, y) \in \mathbb{R}^2$ and $\sigma \in S_2$, the action is given by*

$$\sigma \cdot (x, y) = \begin{cases} (x, y) & \text{if } \sigma = e\,, \\ (y, x) & \text{else}\,. \end{cases}$$

*We equip the output space $\mathbb{R}$ with the trivial (identity) group action. In this setting, the space of equivariant functions is $\mathcal{U} = \{f \in \mathcal{L}(\mathbb{R}^2, \mathbb{R}) : f(x, y) = f(y, x)\,, \ \forall x, y \in \mathbb{R}\}$. Since the output is invariant under the group action, the equivariant functions are, in fact, invariant with respect to the action of $S_2$ on the input space. In the following we identify each linear map $f \in \mathcal{L}(\mathbb{R}^2, \mathbb{R})$ with its matrix representation $w \in \mathbb{R}^2$ given the standard basis, $f(x) = w^\top x$. Therefore $\mathcal{U}$ is a one-dimensional subspace corresponding to the diagonal line in $\mathbb{R}^2$. The average operator $\mathcal{Q} : \mathcal{L}(\mathbb{R}^2, \mathbb{R}) \to \mathcal{U}$ is given by the orthogonal projection*

$$\mathcal{Q}(w) = \left( \frac{w_1 + w_2}{2}, \frac{w_1 + w_2}{2} \right)^\top\,.$$

*Next, let $\mu = \mathcal{N}(0, I_2)$ and $\nu = \mathcal{N}(m, I_2)$ be Gaussian probability distributions on $\mathbb{R}^2$ with $m = (1, 0)^\top$. We first compute the KL divergence directly:*

$$D_{\mathrm{KL}}(\nu \| \mu) = \frac{1}{2} \left( \|m\|^2 + \mathrm{tr}(I_2^{-1} I_2) - 2 - \log \det(I_2^{-1} I_2) \right) = \frac{1}{2}\,.$$

*Next, we compute the pushforward distributions under the linear operator $\mathcal{Q}\mathcal{Q}$. Since $\mathcal{Q}$ is linear, the pushforward measures are given by*

$$\mathcal{Q}_* \mu = \mathcal{N}(0, \Sigma), \quad \mathcal{Q}_* \nu = \mathcal{N}(\mathcal{Q}(m), \Sigma)\,,$$

*where the covariance matrix $\Sigma$ and the mean $\mathcal{Q}(m)$ are given by*

$$\Sigma = M_\mathcal{Q} I_2 M_\mathcal{Q}^\top = \frac{1}{2}\begin{bmatrix} 1 & 1 \\ 1 & 1 \end{bmatrix}, \quad \mathcal{Q}(m) = \left(\frac{1}{2}, \frac{1}{2}\right).$$

*Here, $M_\mathcal{Q} = \begin{bmatrix} \frac{1}{2} & \frac{1}{2} \\ \frac{1}{2} & \frac{1}{2} \end{bmatrix}$ denotes the matrix representation of $\mathcal{Q}$ with respect to the standard basis. We now restrict to the subspace $\mathcal{U}$, and compute the KL divergence:*

$$D_{\mathrm{KL}}(\mathcal{Q}_*\nu \| \mathcal{Q}_*\mu) = \frac{1}{2}\left(\left(\frac{1}{\sqrt{2}}\right)^2\right) = \frac{1}{4}.$$

*To compute the conditional KL divergence, observe that the conditional distributions $\kappa_\mu$ and $\kappa_\nu$ are Gaussians in the orthogonal direction $\mathcal{U}^\perp = \{(x, -x)\}$ with mean 0 and $1/\sqrt{2}$ respectively, and unit variance. Thus,*

$$D_{\mathrm{KL}}(\kappa_\nu(u, \cdot) \| \kappa_\mu(u, \cdot)) = \frac{1}{2}\left(\left(\frac{1}{\sqrt{2}}\right)^2\right) = \frac{1}{4},$$

*for all $u \in \mathcal{U}$. Therefore,*

$$\int_\mathcal{U} D_{\mathrm{KL}}(\kappa_\nu(u, \cdot) \| \kappa_\mu(u, \cdot)) \, \mathcal{Q}_*\nu(du) = \frac{1}{4}.$$

*By Corollary 3.4 in the main paper, we confirm:*

$$D_{\mathrm{KL}}(\nu \| \mu) = D_{\mathrm{KL}}(\mathcal{Q}_*\nu \| \mathcal{Q}_*\mu) + \int_\mathcal{U} D_{\mathrm{KL}}(\kappa_\nu(u, \cdot) \| \kappa_\mu(u, \cdot)) \, \mathcal{Q}_*\nu(du) = \frac{1}{4} + \frac{1}{4} = \frac{1}{2}.$$

*Since in this example the conditional KL divergence is strictly positive, the KL divergence for the projected distributions is strictly smaller than the KL divergence of the initial distributions.*

The reduction of the KL divergence established in the previous example has direct implications for the complexity term appearing in PAC-Bayesian generalization bounds. In particular, since the KL term forms the principal component of the bound's data-independent complexity measure, its decomposition and reduction naturally lead to tighter guarantees. To make this connection explicit, we next illustrate how the reduced KL divergence translates into an improved bound in the setting of McAllester's PAC-Bayes formulation, which serves as the reference bound throughout the main paper.

**Example D.2.** *Consider the same setting as in Example D.1. In this setting, the generalization bound from (10) becomes strictly tighter in the equivariant case. As shown in the previous Example D.1, the KL divergence halves for the equivariant case. Consequently,*

$$\sqrt{\frac{D_{\mathrm{KL}}(\mathcal{Q}_*\mathbb{Q} \| \mathcal{Q}_*\mathbb{P}_H) + \log\frac{1}{\delta} + \log n + 2}{2n - 1}} < \sqrt{\frac{D_{\mathrm{KL}}(\mathbb{Q} \| \mathbb{P}_H) + \log\frac{1}{\delta} + \log n + 2}{2n - 1}}.$$

*This explicit relation illustrates how enforcing equivariance not only reduces the divergence term but also strictly tightens the overall PAC-Bayes bound.*

