# OpenReview forum: "Symmetries in PAC-Bayesian Learning"
_ICML.cc/2026/Conference — ICML 2026 regular_

### Official Review · Reviewer_1sdQ · 2026-03-11

**Soundness:** 3
**Presentation:** 2
**Significance:** 2
**Originality:** 3
**Overall Recommendation:** 4
**Confidence:** 3

**Summary:**

The paper introduces new ways to incorporate symmetries in both the hypothesis space and the data into PAC-Bayesian generalization bounds for possibly non-compact symmetry groups. In particular, the authors show that accounting for symmetries in the hypothesis space can lead to tighter generalization bounds by reducing the KL-divergence compared to bounds that ignore these symmetries. Furthermore, under the additional assumption that the loss function is convex and
$\mathcal{Q}$-invariant, the authors derive tighter generalization guarantees even when the data distribution itself is not invariant to the considered symmetries.

**Compliance With Llm Reviewing Policy:**

Affirmed.

**Final Justification:**

The authors provided some clarifications; however, I believe the paper would benefit from more precision regarding the Neural Network training. I maintain my score as a weak accept.

**Key Questions For Authors:**

1. The theoretical guarantees rely on the hypothesis class being constructed to respect the symmetries exhibited in the data distribution, in practice how restrictive on the hypotheses is this assumption ?

2. In the experimental section the optimization of the posterior distribution is done in a self-bounding fashion using McAllester bound, in the case of the equivariant CNN is it the proposed bound that is used (Theorem 4.3, Equation (14)) ? Furthermore, is the bound suitable for self-bounding or does it imply more difficulties than the classical McAllester one ?

3. For the training of the models the cross-entropy loss is used, and the 0-1 loss is considered for evaluation, are they both $\mathcal{G}$-invariant as required by assumption 4.1 ?

4. In Remark 3.5 you say "The mapping Q acts as a (possibly lossy) transformation of hypotheses" can you clarify what you mean by "lossy" in this case ?

**Limitations:**

The authors partially discuss the limitations of their approach, particularly the reliance on the assumption that the hypothesis class respect the symmetry of the data. It could be beneficial to discuss conditions for this assumption to be verified. Given the theoretical nature of the work, no direct societal risks are apparent.

**Strengths And Weaknesses:**

**Soundness.** Overall, the claims of the paper are well supported. The theoretical results originate from well-established PAC-Bayesian theory and the adaptations to symmetries are sound. Some assumptions may be strong, such as Assumption 4.1, which requires the loss to be $\mathcal{G}$-invariant. While this may be justifiable, it raises the question of whether standard losses, such as the 0-1 loss, satisfy this assumption, and whether it affects the computation of the bound. Moreover, the main results rely on the hypothesis class and the data distribution sharing the same symmetries, which may be difficult to satisfy in practice.

**Presentation.** The paper is generally well structured but some parts are difficult to follow, mainly due to notation and the introduction of key concepts without sufficient intuition. In particular, the notation $\mathcal{Q}*Q$ used for the symmetrized posterior is introduced without a clear explanation of its construction or interpretation. This makes it difficult to understand the role of this object. Providing a more explicit definition and an intuitive explanation or example would significantly improve readability. The same remark can be applied to $\pi_{\mathcal{X}_{\phi}}$ and $\pi_{\mathcal{G}}$. Aside from these issues, the narrative is coherent and the paper situates itself reasonably well within the PAC-Bayesian and equivariant learning literature.

**Significance.** The paper addresses the problem of symmetries in models and data introduces new ways of taking the hypothesis class into account in the PAC-Bayesian setting, which is relevant for some applications of machine learning such as computer vision. It deepens the theoretical understanding of the subject to non-compact groups of symmetries and non-invariant data distributions. From the PAC-Bayesian standpoint, it introduces new ways of taking into consideration the hypothesis class. However, considering symmetries in the hypothesis class, while it leads to tighter bounds, may also lead to larger risks by itself, and the assumption that the hypothesis and data show the same symmetries is necessary to overcome this limitation but may be restrictive.

**Originality.** The work provides new theoretical insights by analyzing the impact of symmetries on the KL-divergence and risk when they are properly accounted for in both the hypothesis class and the data distribution. While it builds on standard PAC-Bayesian results, the paper refines them to handle symmetries even when the symmetry group is non-compact.

---

> ### Author Rebuttal · Authors · 2026-03-31
>
> 1. We agree that requiring the hypothesis class to reflect the symmetries of the data distribution is a nontrivial modeling assumption. However, this is not overly restrictive in practice and a standard and well-established design principle in modern machine learning. Indeed, many widely used architectures are explicitly built to encode symmetries like CNNs encoding translation equivariance, Group-equivariant CNNs and steerable CNNs encoding rotations and other group actions and also Transformers incorporating permutation invariance/equivariance in certain components.
> Our contribution provides theoretical guarantees from a PAC-Bayesian point of view, that those approaches that include symmetries guarantee better results.
>
> 2. Yes, in the equivariant CNN experiments we use the the bound derived in Theorem 4.3 (Eq. 14).
> Regarding the “self-bounding” aspect, our bound is fully suitable for this type of optimization. In particular, it has the same structural form as standard PAC-Bayesian objectives, so it can be minimized with respect to the posterior in exactly the same way.     Importantly, the proposed bound does not introduce additional optimization difficulties compared to the classical McAllester bound. We will clarify this in the experimental section.
>
> 3. Yes, both losses satisfy Assumption 4.1 in our setting, due to how the group acts on inputs and labels.
>  In our experiments, The group (rotations/translations) acts on inputs $x$, while the labels $y$ remain unchanged, i.e., the group acts trivially on the label space. Therefore, for any transformation $g$ and label $y$ it holds that $g \cdot y = y$.
> As a consequence, Assumption 4.1 $\ell (g \cdot y_1, g \cdot y_2) = \ell (y_1, y_2)$ holds for all transformations $g$ and labels $y_1, y_2$. More generally, Assumption 4.1 holds when the group action preserves labels (standard in classification tasks) for any loss function.
>
> 4. We thank the reviewer for pointing out that this terminology was unclear.
> By “lossy”, we mean that the transformation does not guarantee improvement in risk, when the symmetries in the data don't fit. Only with the theory in section 4, this can be guaranteed. This problem ois discussed in more detail in the second part of Remark 3.7. We agree that this point was introduced too early without sufficient explanation. In the revision, we will remove the term “lossy” at that point, since a more detailed explanation is given in the following Remark 3.7
>
> **Presentation:**
> We thank the reviewer for highlighting these presentation issues.
> For the symmetrized posterior $\mathcal{Q}_* \mathbb{Q}$, we will provide a more explicit definition of the symmetrized posterior as a pushforward induced by the averaging operator, and complement this with a concrete example. In particular, we will extend the example in Appendix C to not only illustrate the reduction of the KL term, but also to give an intuitive understanding of how hypotheses are transformed under symmetrization.
>
> We thank the reviewer for his suggestions. To improve the understanding how the averaging operator is used as a pushforward to symmetrize the hypotheses, we will add an explicit example, In particular, we will expand the example in Appendix C, to not only illustrate the reduction of the KL term, but also to give an intuitive understanding of how hypotheses are transformed under symmetrization.
> We agree that the notation $\pi_{\mathcal{X}{\phi}}$ and $\pi{\mathcal{G}}$ is currently difficult to interpret. we will illustrate the projections in the case of the rotation group acting on $\mathbb{R}^2$, where the construction reduces to polar coordinates. This provides an intuitive and familiar interpretation that helps relate the abstract operators to standard concepts.

---

> > ### Author Rebuttal · Reviewer_1sdQ · 2026-04-02
> >
> > About answers 1 and 2, since your predictor is defined as a neural network, can you provide more details about the way you optimize it? Training neural networks is a non-trivial problem in the PAC-Bayesian framework, and the paper could benefit from additional clarification on this matter.
> >
> > Specifically, in Equation 3 you define an expectation over all functions f, which corresponds to the quantity you later bound: the risk of the expectation over all classifiers drawn according to the distribution Q. While this is a standard definition of risk in PAC-Bayesian theory (the Gibbs risk), it is not immediately applicable to the training and bounding of a single neural network, and more details are needed to clarify this point.
> >
> > In particular, you define your hypotheses to be G-equivariant. Does this mean that each hypothesis corresponds to a neural network, in which case you learn multiple networks simultaneously, or does the entire hypothesis set encode a single network?
> >
> > Furthermore, you mention that encoding symmetries is widely used in CNNs and transformers. Can you provide references supporting this claim? Is it also possible to define a hypothesis set that encodes symmetries without relying on neural networks, so that your bound can be applied more directly?

---

> > > ### Author Response · Authors · 2026-04-07
> > >
> > > In our work, we use isometric gaussians for the prior and posterior. We train a single neural network by minimizing the empirical risk. Concretely, the network’s parameters correspond to the mean $\mu$ of the posterior distribution $\mathbb{Q}$. The variance $\sigma^2$ of $\mathbb{Q}$ is then chosen to minimize the right-hand side of the PAC-Bayesian bound, which can be done either via grid search or optimization. This approach allows us to directly compare our symmetry-aware bound with the classical McAllester bound.
> > >
> > > In the PAC-Bayesian setting, the bound controls the Gibbs risk, i.e., the expected risk of predictors drawn from the posterior $\mathbb{Q}$, rather than the risk of a single deterministic network. In our setting, $\mathbb{Q}$ is defined as an isotropic Gaussian $\mathcal{N}(\mu, \sigma^2)$ over the neural network parameters.
> > > The mean $\mu$ can be interpreted as the deterministic network we train, while the variance $\sigma^2$ introduces a notion of stochasticity that can also be understood as modeling the randomness inherent in SGD or its variants. The expectation in Equation (3), i.e. the Gibbs risk, is approximated in practice using Monte Carlo sampling from $\mathbb{Q}$.
> > >
> > >
> > > Convolutional neural networks implement translation invariance via weight sharing, the same filter is used for every spatial location (see Yann LeCun et al., Gradient-Based Learning Applied to Document Recognition). This idea is generalized to arbitrary group actions in group-equivariant CNNs by Taco Cohen and Max Welling in "Group equivariant convolutional networks".
> > > In transformers, the self-attention mechanism is permutation equivariant with respect to the input sequence, before positional encodings are added (Ashish Vaswani et al., Attention Is All You Need). Variants such as Set Transformers (Lee et al. "Set Transformer: A Framework for Attention-based Permutation-Invariant Neural Networks") explicitly enforce permutation invariance.
> > > Finally, our framework is not restricted to neural networks. The hypothesis class can, in principle, consist of any family of functions satisfying the required symmetry constraints, e.g. invariant kernel methods. Neural networks are simply a convenient and expressive instantiation used in our experiments.

---

### Official Review · Reviewer_82km · 2026-03-13

**Soundness:** 3
**Presentation:** 3
**Significance:** 3
**Originality:** 2
**Overall Recommendation:** 4
**Confidence:** 4

**Summary:**

The authors provide a tightening of McAllester’s PAC-Bayes bound in the case on non-compact groups and non-invariant probability distributions. They verify their theory in the case of rotated and translated MNIST trained on both a CNN and an E2-CNN and compare the original to the tighter symmetry-aware bound.

**Compliance With Llm Reviewing Policy:**

Affirmed.

**Ethical Review Concerns:**

In the Supplementary Material there are multiple references to AISTATS 2026. If this paper was retracted from AISTATS this is no cause for concern. From the deadlines it seems like publishing at both might count as a double submission, which needs looking into.

**Ethics Expertise Needed:**

["Research Integrity Issues (e.g., plagiarism)"]

**Final Justification:**

The authors added a sanitized 2D and CIFAR experiment. They discussed the limitation in the suggested format, namely by letting go of each assumption needed for the theorem in the paper. I raised my score to a weak accept.

**Key Questions For Authors:**

- Can you show the performance of the newly derived bound on simple mathematical data with known symmetries as well? This would be nice to showcase the performance of the bound in a sanitised setting before moving on to handwritten digits.
- How about other datasets beyond rotated/translated MNIST? Would this work for CIFAR, physical data (trajectories), or other types of data? (I.e., is there a computational issue for images larger than MNIST?)
- Does the bound work for more sophisticated distributions and transformations, such as scaling, or non-affine transformations?
- Where does the improvement in this new bound come from, explicitly? Is an analysis of this improvement possible?
- How does this work connect to the two papers below, are they relevant?

"A PAC-Bayesian Generalization Bound for Equivariant Networks", Behboodi et al. (NeurIPS 2022)

"PAC-Bayes Compression Bounds So Tight That They Can Explain Generalization", Lotfi et al. (NeurIPS 2022)

**Limitations:**

A thorough discussion regarding limitations of this bound is sorely missing. E.g., What happens when one of the assumptions does not hold?

**Strengths And Weaknesses:**

Strengths:
1. The paper provides proofs of a theorem for a symmetry-aware bound in the case of data distributed along orbits for non-compact groups and non-invariant distributions.
2. The discussion tackles a valid subfield of interest, namely the non-compact groups and non-invariant distributions of data along orbits.
3. The assumptions are well laid out and the theorem seems to yield strong improvements on the original bound.

Weaknesses:
1. Despite the extensive theoretical exposition, the experimental section is missing other settings beyond MNIST or other models. Unfortunately the breadth of experiments is too narrow to evaluate the accuracy, usefulness, or flexibility of the derived bound.
2. Only rotation and translation are considered. Surely there are more transformations for which this would hold or experimental evidence would be interesting.
3. The paper does not compare to other possible bounds for symmetrically distributed data. Some highly relevant published papers that discuss PAC-Bayes for DL or even in the context of symmetry were not referenced or used as baselines. (See below.)

"A PAC-Bayesian Generalization Bound for Equivariant Networks", Behboodi et al. (NeurIPS 2022)

"PAC-Bayes Compression Bounds So Tight That They Can Explain Generalization", Lotfi et al. (NeurIPS 2022)

---

> ### Author Rebuttal · Authors · 2026-03-31
>
> 1. We agree that a controlled setting is valuable to isolate the effect of symmetry. In fact, our theory admits explicit analysis in low-dimensional settings, which we include in Appendix C. In a simple 2D example with a symmetry given by the symmetric group $S_2$, we show analytically that symmetrization reduces the KL divergence by a factor of 2. This directly translates into a tighter PAC-Bayes bound.
>
> 2. In addition to the results on MNIST in the submission, we have conducted further experiments on CIFAR-10 using steerable CNNs (via the e2cnn library) as for the MNIST experiment. We observe consistent improvements aligned with our theory:
> * Baseline (not equivariant):
> Mean loss: 0.7762 ± 0.0143
> KL(Q||P) = 6330.988312, complexity term = 0.325180, McAllester bound = 1.095443
> * Equivariant:
> Mean loss: 0.6997 ± 0.0083
> KL(Q||P) = 4696.081543, complexity term = 0.280165, McAllester bound = 0.972208
> These results confirm that symmetry-aware priors reduce the KL term and improve the bound in practice, beyond the rotated-MNIST setup. Thus, the approach is not limited to small images and is applicable to larger-scale settings.
>
> 3. Our framework is intentionally general. The bound makes no assumption on the distribution, so the bound works also for more sophisticated distributions. In practice, we use gaussians distributions, since they admit closed-form KL expressions and are standard in PAC-Bayes deep-learning. Other choices (e.g., Laplace, mixtures) are equally compatible. Regarding the transformations, the only restriction is, that the transformation is given by a group action. Hence, the bound also works for other transformations including scaling and non-affine transformation.
>
> 4. The improved tightness of the bound comes from the reduction of the KL divergence between the symmetrized prior and posterior in Lemma 3.4. Reducing the KL divergence implies a reduction of the complexity term and hence a tighter bound. Further Lemma 3.4 provides a decomposition of the KL divergence into the KL divergence between the symmetrized distributions and a rest term. To analyze the improvement of the bound, one has to analyze the rest term. We did this for the simple two dimensional case with gaussian distributions in Appendix C. In this example the KL divergence reduces by a factor of $2$.
>
> 5. We thank the reviewer for pointing out these relevant works and agree they should be discussed.
> * "A PAC-Bayesian Generalization Bound for Equivariant Networks", Behboodi et al. (NeurIPS 2022):
> The main result in the paper is a generalization bound for a specific instant of an equivariant multilayer perceptron instead of a posterior distribution, which is the standard PAC-Bayesian approach.
> The difference to our contribution is, that we do not focus on multilayer perceptrons but on general hypotheses. We provide an typical PAC-Bayesian bound that evaluates the risk and empirical risk with respect to a posterior. Further, our approach ensures that for models which respect the symmetry inherent in the data, this symmetry leads to lower risks and not only tighter bounds.
>
> * "PAC-Bayes Compression Bounds So Tight That They Can Explain Generalization", Lotfi et al. (NeurIPS 2022):
> Their compression-based approach differs fundamentally from ours. Their method improves Catoni's bound by carefully designing a prior and posterior tailored to a compressed network. Our work, by contrast, does not impose such restrictions. It provides a general bound that holds for arbitrary priors and posteriors, and leverages inherent data symmetries to achieve tighter guarantees.
>
> **Limitations:**
> We agree that the limitations should be stated more clearly. The main limitation is that requiring the hypothesis class to reflect the symmetries of the data distribution is a nontrivial modeling assumption. However, this is not overly restrictive in practice. It is a well-established design principle in modern machine learning.
>
> If one of the assumptions is dropped our theoretical results no longer hold. If Assumption 4.1 is not satisfied, it is possible to construct counter examples. However this assumption is mild, it holds trivially whenever the group action preserves labels, which is standard in classification tasks for any loss function. Assumption 3.1 is also mild, and satisfied in almost every application. Assumption 2.1 formalizes the presence of symmetry in the data. Without it, there are no symmetries to exploit, rendering the entire approach inapplicable by definition.
>
> **Ethical Review Concerns:**
> Thank you for pointing this out.
>
> The work was submitted to AISTATS but rejected before the ICML submission deadline. The ICML submission is therefore not concurrent. After rejection, we revised the theoretical core before submitting to ICML. The remaining AISTATS references in the supplementary material (README) were an oversight and will be removed in the final version.

---

> > ### Author Rebuttal · Reviewer_82km · 2026-04-03
> >
> > I thank the authors for their clarifications and the answers to my questions.
> >
> > - (1, 2) Adding the sanitized examples to the paper would really strengthen it and validatide the theorem in stricter settings. I also appreciate the extra experiment on CIFAR data and glad to see it also confirms your results.
> >
> > - (3) Having even more transformations would strengthen the paper further. I would highly recommend looking into these extension for future work.
> >
> > - (4, 5) Thank you for clarifying. Same for the limitations, this breakdown would be nice to see discussed in the paper.
> >
> > I have increased my score.

---

> > > ### Author Response · Authors · 2026-04-07
> > >
> > > We thank the reviewer for their thoughtful follow-up and for increasing their score after considering our rebuttal. We greatly appreciate your careful evaluation and constructive feedback throughout the process.

---

### Official Review · Reviewer_pehs · 2026-03-18

**Soundness:** 3
**Presentation:** 3
**Significance:** 3
**Originality:** 3
**Overall Recommendation:** 4
**Confidence:** 3

**Summary:**

The paper studies PAC-bayes generalization bounds for learning under symmetry constraints. Specifically, it extends existing PAC-Bayes analyses to non-compact groups and non-invariant data distributions. The authors modify classic McAllester's PAC-Bayes bounds to incorporate data-dependent and non-uniform priors that align with non-compact group actions. In addition, they validate the theory on rotated-MNIST experiments with non-uniform rotation distribution, and demonstrate improvement compared to symmetry-agnostic baselines, in line with their theory.

**Compliance With Llm Reviewing Policy:**

Affirmed.

**Final Justification:**

I thank the authors for their response.

Most of my questions have been addressed adequately. Some of my points remain, particularly having more extensive experiments, since this seems a contribution, could make the empirical claims of the paper stronger. In relation to these experiments, it would be interesting to have more transformations, particularly because having a more general theory (e.g. non-compact groups) appears to be one of the paper’s main benefits.

I retain my score recommending acceptance.

**Key Questions For Authors:**

Some questions in relation to weaknesses mentioned above.

- Would the authors agree that the novelty or the derived extensions of the existing PAC‑Bayes bounds are, in themselves, relatively modest? (To be clear, I am not asking this because I would find this reason for rejection, since careful and concise extensions can still be highly valuable to the community. However, clarification here would help me better understand the intended contribution.) Conversely, if the extensions by itself are deemed conceptually significant or technically nontrivial, it would be helpful if authors could highlight these steps more explicitly, both to guide reader/reviewers and strengthen the presentation of the theoretical advances.
- Are there reasons why additional experiments (G-CNN, Steerable CNN, more groups, deeper architectures, etc..) would not be relevant for the theory?
- Are there situations where symmetric priors hurt bound tightness?
- Can we say anything on how the approach scales to larger/deeper architectures? (e.g. effect on bound tightness)

**Limitations:**

-

**Strengths And Weaknesses:**

Strengths:
- Symmetries are a very most important types of inductive bias that are often incorporated in machine learning / deep learning architectures (e.g. group-convolutional layers) to improve generalization and learning efficiency. Providing theoretic guarantees on the generalization properties, as done in this work, provides an improtant rigorous foundation behind such techniques, which have been found to work well empirically.
- Prior PAC‑Bayesian treatments of symmetry (e.g. Lyle et al and Elesedy & Zaidi) require compact groups and invariant data distributions. Extending to non-compact groups further extends theory.

Weaknesses:
- While extending to non‑compact groups is conceptually meaningful, the adaption of the McAllester’s bound feels technically straightforward. It is less clear whether theoretical innovations go beyond careful bookkeeping. Perhaps I have not fully grasped or missed certain complexity in the contribution, and even if contribution itself is not hard, I expect that there is value...
- The paper presents only a single relatively small‑scale experiment. It would be interesting to extend this with more baselines (standard CNN, G-CNN, steerable CNNs). Or are there reasons why this would not be relevant for the theory?
- Related to the above, it would be interesting to have some more discussion on bound tightness in practice.

---

> ### Author Rebuttal · Authors · 2026-03-31
>
> 1. We agree that the main idea to use PAC-Bayesian Bounds and prove a reduction in the KL term is not new. However, we would like to clarify that the main contribution is not a direct extension by bookkeeping, but rather enabling PAC-Bayes analysis in a regime where existing tools break. Concretely, prior works rely on the compactness of the group and data invariance. Our setting removes both assumptions, which comes with nontrivial technical challenges.
> Without compactness, Haar measures are not finite, so standard averaging arguments are ill-defined. Further do standard averaging arguments require invariant data distributions, as the Haar measure is invariant. We resolve this by introducing a data-dependent averaging operator, which replaces Haar integration.
> The key step is Theorem 4.3, where we show that symmetrizing the hypotheses using the new averaging operator leads to improvements in the true risk as well as in the bound itself by tightening it.
> We will revise the paper to make these points more explicit and highlight precisely where standard arguments fail and how our construction overcomes this.
>
> 2. We agree that broader empirical validation would strengthen the paper. In addition to the results on MNIST in the submission, we have conducted further experiments on CIFAR-10 using steerable CNNs (via the e2cnn library) as for the MNIST experiment. We observe consistent improvements aligned with our theory:
> * Baseline (not equivariant):
> Mean loss: 0.7762 ± 0.0143
> KL(Q||P) = 6330.988312, complexity term = 0.325180, McAllester bound = 1.095443
> * Equivariant:
> Mean loss: 0.6997 ± 0.0083
> KL(Q||P) = 4696.081543, complexity term = 0.280165, McAllester bound = 0.972208
> These results confirm that symmetry-aware priors reduce the KL term and improve the bound in practice, beyond the rotated-MNIST setup.
> Finally, the theory predicts that improvements become more pronounced when the group is larger or richer, or multiple symmetries are present. This follows from Lemma 3.4, where the KL divergence is decomposed into a Kl divergence between push-forwards and a rest term.
>
> 3. A symmetric prior does not loosen the bound. However, if the imposed symmetry is mismatched with the symmetry in the data, then the true risk and the empirical risk will most likely both increase. The performance of the learned symmetric posterior will be worse.
>
>
> 4. Deeper architectures are interesting from the application point of view. From a PAC-Bayesian perspective, the key quantity is the KL divergence relative to sample size, not depth per se. For example for posterior prior that are isotropic Gaussians ($\mathbb{P} = \mathcal{N} (\mu_p, \sigma_p^2 I)$ and $\mathbb{Q} = \mathcal{N} (\mu_p, \sigma_p^2 I)$ on $\mathbb{R}^d$), the KL divergence is
>     \begin{align*}
>         \frac{1}{2}\left(d \ln{\frac{\sigma_q^2}{\sigma_p^2}} - d + d\frac{\sigma_p^2}{\sigma_q^2} + \frac{1}{\sigma^2_q} \|\mu_q - \mu_p\|^2 \right),
>     \end{align*}
> which scales linear in the dimension d. Thus the bound remains tight, if the sample size $n$ grows proportionally to the model size $d$.

---

> > ### Author Rebuttal · Reviewer_pehs · 2026-04-07
> >
> > I thank the authors for their response.
> >
> > Most of my questions have been addressed adequately. Some of my points remain, particularly having more extensive experiments, since this seems a contribution, could make the empirical claims of the paper stronger. In relation to these experiments, it would be interesting to have more transformations, particularly because having a more general theory (e.g. non-compact groups) appears to be one of the paper’s main benefits.
> >
> > I retain my score recommending acceptance.

---

> > > ### Author Response · Authors · 2026-04-08
> > >
> > > We thank the reviewer for the constructive feedback and support.
> > > We agree that additional datasets and a broader range of transformations could further strengthen the empirical section. We will incorporate the CIFAR experiment into the final version and expand the experimental section where feasible.

---

### Decision · Program_Chairs · 2026-04-30

**Decision:**

Accept (regular)

**Comment:**

The authors study PAC-Bayesian generalization bounds for learning under symmetries, in the setting where the data distribution is not necessarily invariant and the group may be non-compact.

All reviewers find the theoretical results in the paper interesting and well motivated, well related to the ML community, and they appreciate the non-compact setting. They all provided (weak) accept scores, and they did not strongly argue against acceptance.

I found this paper very interesting too, and I think the authors could successfully take care of the reviewers’ comments; thus, I recommend acceptance.

That being said, it would be excellent if the authors could add more extensive experiments to their paper for the sake of completeness:

> Reviewer pehs: Some of my points remain, particularly having more extensive experiments, since this seems a contribution, could make the empirical claims of the paper stronger. In relation to these experiments, it would be interesting to have more transformations, particularly because having a more general theory (e.g. non-compact groups) appears to be one of the paper’s main benefits.


> Reviewer 1sdQ: I believe the paper would benefit from more precision regarding the Neural Network training.


I ask the authors to carefully consider the application of these comments in the next version of their paper, as my recommendation is conditional on successfully applying the reviewers’ comments and discussion in the next version.

Note: For the next version, please remove the reference to AISTATS